



# Evaluation of the WMO-SPICE transfer functions for adjusting the wind bias in solid precipitation measurements

Craig D. Smith[1], Amber Ross[1,2], John Kochendorfer[3], Michael E. Earle[4], Mareile Wolff[5], Samuel Buisan[6], Yves-Alain Roulet[7], Timo Laine[8]

[1]Climate Research Division, Environment and Climate Change Canada, Saskatoon, S7N 3H5, Canada
[2]Global Institute for Water Security, University of Saskatchewan, Saskatoon, S7N 3H5, Canada
[3]Atmospheric Turbulence and Diffusion Division, ARL, National Oceanic and Atmospheric Administration, Oak Ridge, 37830, USA
[4]Meteorological Service of Canada, Environment and Climate Change Canada, Dartmouth, B2Y 2N6, Canada
[5]Norwegian Meteorological Institute, Oslo, 0313, Norway
[6]Delegación Territorial de AEMET (Spanish National Meteorological Agency) en Aragon, Zaragoza, 50007, Spain
[7]MeteoSwiss, Payerne, 1530, Switzerland
[8]Finnish Meteorological Institute, Helsinki, 00101, Finland

*Correspondence to:* Craig D. Smith (craig.smith2@canada.ca)

**Abstract.** The World Meteorological Organization (WMO) Solid Precipitation Inter-Comparison Experiment (SPICE) involved extensive field intercomparisons of automated instruments for measuring snow during the 2013/2014 and 2014/2015 winter seasons. A key outcome of SPICE was the development of transfer functions for the wind bias adjustment of solid precipitation measurements using various precipitation gauge and windshield configurations. Due to the short intercomparison period, the dataset was not sufficiently large to develop and evaluate transfer functions using independent precipitation measurements. The present analysis uses data collected at eight SPICE sites over the 2015/2016 and 2016/2017 winter periods, comparing 30-minute adjusted and unadjusted measurements from Geonor T-200B3 and OTT Pluvio$^2$ precipitation gauges in different shield configurations to the WMO Double Fence Automated Reference (DFAR) for the verification of the transfer function. Performance is assessed in terms of relative total catch (RTC), root mean square error (RMSE), Pearson correlation (r), and Nash-Sutcliffe Efficiency (NSE) for all precipitation types, and for snow only. The evaluation shows that the performance varies substantially by site. Adjusted RTC varies from 54% to 123%, RMSE from 0.07 mm to 0.38 mm, r from 0.28 to 0.94 and NSE from -1.88 to 0.89, depending on precipitation phase, site, and gauge configuration. Generally, windier sites such as Haukeliseter (Norway) and Bratt's Lake (Canada) exhibit a net under-adjustment (17% to 46%), while the less windy sites such as Sodankylä (Finland) and Caribou Creek (Canada) exhibit a net over-adjustment (2% to 23%). Although the application of transfer functions is necessary to mitigate wind bias in solid precipitation measurements, especially at windy sites and for unshielded gauges, the inconsistency in the performance metrics among sites suggests that the functions be applied with caution.

## 1 Introduction

The World Meteorological Organization (WMO) Solid Precipitation Inter-Comparison Experiment (SPICE) was a Commission for Instruments and Methods of Observation (CIMO) initiative to assess and compare instruments and methods for measuring solid precipitation (Nitu et al., 2012; Nitu et al., 2018). The objectives were: 1) to make



recommendations for appropriate automated field reference systems; and 2) to provide guidance on the performance and operation of automated systems for measuring solid precipitation and snow on the ground. SPICE was motivated by the need for accurate and homogenized solid precipitation measurements. For example, such measurements are required for climate trend analysis in northern regions (e.g. Førland and Hanssen-Bauer, 2000; Yang and Ohata, 2001,

Scaff et al., 2015). Following historical works on adjusting the systematic undercatch of solid precipitation measurements due to wind (Goodison, 1978; Sevruk et al., 1991; Goodison et al., 1998; Sevruk et al., 2009; Smith, 2009; Wolff et al., 2015; Kochendorfer et al., 2017a; Buisan et al., 2017), a methodology and a set of widely applicable transfer functions for the adjustment of high resolution (i.e. 30-min) precipitation measurements was developed. The SPICE transfer functions discussed in this study were developed for single-Alter-shielded or unshielded automated

precipitation gauges (Kochendorfer et al., 2017b). Because of the symbioses between the Kochendorfer et al. (2017b) SPICE work and this evaluation, the SPICE methodology is described below in more detail and henceforth cited as K2017b.

The official SPICE intercomparison period occurred during the winters of 2013/2014 and 2014/2015. During this

period, a Double Fence Automated Reference (DFAR) was operated at eight test bed sites: Bratt's Lake (XBK), Caribou Creek (CCR), the Centre for Atmospheric Research and Experiments (CARE; abbreviated to CAR), Formigal (FOR), Haukeliseter (HKL), Marshall (MAR), Sodankylä (SOD), and Weissfluhjoch (WFJ) (Fig. 1; Table 1). The DFAR was developed for use as the field reference configuration for SPICE (Nitu, 2012; Nitu et al., 2016; Nitu et al., 2018). A detailed layout and site description for each of these eight sites can be found in the WMO-SPICE site

commissioning reports at: http://www.wmo.int/pages/prog/www/IMOP/intercomparisons/SPICE/SPICE.html and in the WMO-SPICE final report (IMO 131 found at http://www.wmo.int/pages/prog/www/IMOP/publications-IOM-series.html; Nitu et al., 2018).

The DFAR consisted of either a Geonor T-200B3 or an OTT Pluvio$^2$ automated weighing precipitation gauge with a

single-Alter shield inside the same large, octagonal double fence used for the WMO Double Fence Intercomparison Reference (DFIR). The DFIR was employed as a manual field reference configuration during previous solid precipitation intercomparisons (Yang et al., 1993; Goodison et al., 1998). The DFAR incorporated a precipitation detector as a verification tool to reduce the probability of using false precipitation reports in SPICE data analysis. The result was the development of a high confidence reference precipitation data set called the Site Event Data Set (SEDS;

Reverdin, 2016). In the SEDS, a precipitation event was a 30-minute period during which the precipitation detector observed precipitation for at least 18 minutes during the 30-minute period (60% of event duration) and the DFAR measured ≥ 0.25 mm. The justification for the filtering criteria used in K2017b is detailed in Kochendorfer et al. (2017a). The SEDS was then used to produce the SPICE transfer functions. By combining data from each site in Table 1, the intent was to make these multi-site transfer functions universally applicable.



## 1.1 The K2017b transfer functions

The transfer functions presented in K2017b were developed for both unshielded and single-Alter shielded automated

precipitation gauges by combining observations from the Geonor T-200B3 and OTT Pluvio$^2$ gauges (hereafter referred to as the sensors under test, or SUT), after demonstrating that the unshielded catch from both SUT types were very similar. Using the SEDS, further processing was applied to the SUT data (as justified in Kochendorfer et al., 2017a) using a minimum 30-minute threshold defined as the median of the ratio of the SUT accumulation to that of the DFAR, for the precipitation event threshold of 0.25 mm. This prevented the results from becoming biased toward the gauge

used in the SEDS event selection. For wind speed, K2017b used the wind speed measurements available at each site and applied the log-profile law to produce a 30-minute average wind speed for both gauge height ($U_{gh}$) and the standard 10 m height ($U_{10m}$), either incrementing speeds to $U_{10m}$ or decreasing speeds to $U_{gh}$, depending on which measured wind height speed data was deemed the best at each site. Catch efficiencies ($C_E$) were then calculated for each event as the ratio of the 30-minute SUT precipitation accumulation to that from the DFAR over the same period. From

K2017b, two functional forms were used to fit the data:

$$C_E = e^{-a(U)(1-\tan^{-1}(b(T_{air}))+c)},\qquad(1)$$

and

$$C_E = (a)e^{-b(U)} + c,\qquad(2)$$

where U is wind speed (specifically either $U_{gh}$ or $U_{10m}$) in m s$^{-1}$, $T_{air}$ is air temperature in degrees C, and *a*, *b*, and *c* are coefficients to fit the data to the model. Eq. 1 and Eq. 2 listed here are referred to as Eq. 3 and Eq. 4 in K2017b.

To reduce the impact of fewer events at higher wind speeds and the potential impacts of blowing snow, the SEDS data was filtered further to remove events with wind speeds higher than 7.2 m s$^{-1}$ (9 m s$^{-1}$) at gauge height (10 m).

The key difference between Eq. 1 and Eq. 2 is the inclusion of temperature dependency in Eq. 1. This allows for a continuous (3-dimensional) transfer function at all temperatures without having explicit knowledge of the precipitation

phase. This curve is shown in Fig. 2 (red) for both the single-Alter shielded (solid) and unshielded (dashed) gauges and a temperature of -5 °C. Equation 2, however, requires an assessment of precipitation phase (liquid, solid, or mixed), with each phase having unique coefficients. K2017b used temperature to discriminate phase for Eq. 2 and assumed the following: solid precipitation occurs at $T_{air} < -2$ °C; liquid precipitation occurs at $T_{air} > 2$ °C; and mixed precipitation occurs at $-2$ °C $\leq T_{air} \leq 2$ °C. Equation 2 is plotted in Fig. 2 (blue) using the coefficients for snow for the

single-Alter shielded (solid) and unshielded (dashed) gauges. For both equations and gauge configurations, unique coefficients were derived for each of the two wind speed measurement heights. The SUT precipitation were adjusted using both Eq. 1 and Eq. 2 by employing the appropriate coefficients, depending on phase/temperature, wind



measurement height, and shield configuration. For Eq. 2, precipitation classified as rain ($T_{air} > 2$ °C) assumed a $C_E$ equal to 1. The coefficients used in Eq. 1 and Eq. 2 are detailed in K2017b and are not repeated here.

## 1.2 K2017b results

Following the end of the SPICE project, the performance of the SPICE transfer functions was evaluated as described in K2017b, using the same SEDS data used to develop those transfer functions. As discussed above, the data from all eight sites in Table 1, including data from multiple gauges of the same configuration at each of the sites, were combined to fit the transfer function models. The evaluation of the universal transfer function equations was completed

for each individual site, arguably making the evaluation at least marginally independent from the pooled, multi-site measurements used to develop the transfer functions. The K2017b results were based on four metrics: the root mean square error (RMSE), mean bias, Pearson correlation (r), and percentage of events (PE) between the DFAR and SUT that agreed within a specified threshold (typically 0.1 mm). It should also be noted that the overall performance metrics summarized in K2017b included all precipitation phases, whether adjusted by the transfer functions (i.e. solid and

mixed) or not (i.e. liquid).

K2017b showed that the SPICE transfer functions reduced the overall bias in the unshielded precipitation gauges (both Geonor T-200B3 and OTT Pluvio[2], all sites combined) from -33.4% to 1.1%, but the results varied by site, with CAR and WFJ showing over-adjustment and HKL, FOR, and XBK showing under-adjustment. For the most part, the r,

RMSE, and PE were slightly improved after adjustment, but these also varied by site. K2017b also showed that, in general, the mountainous sites experienced larger errors after adjustment, with one mountainous site (WFJ) being over-adjusted and the other two (HKL and FOR) being under-adjusted.

## 1.3 Motivation for the extended evaluation

The impetus for this extended evaluation was twofold:

   1)   The methodology used during SPICE for developing and evaluating the transfer functions used only a subset of the observed data (the SEDS), and although this was a robust methodology for developing transfer functions, it did not provide a comprehensive evaluation of the adjustments under more natural

circumstances. This evaluation will use winter precipitation data as they would be collected in an operational or monitoring application, produce time series of adjusted winter precipitation, and evaluate the performance of the adjustment relative to the reference.

   2)   The dataset used for the evaluation of transfer functions in K2017b was not completely independent of that

used to develop the functions. The present evaluation will use two additional years of data collected following the end of the SPICE intercomparison period, and hence will provide an independent means of assessment.



This evaluation will examine the performance of the SPICE transfer functions for precipitation measurements from each of the eight intercomparison sites shown in Fig. 1 for the 2015/2016 and 2016/2017 winter seasons. Each of these sites continued to operate a DFAR following the SPICE intercomparison period, a critical component for assessing transfer function performance. The assessment will be conducted for both unshielded and single-Alter shielded gauges using wind speed heights of 10 m and gauge height, where available. Extending the K2017b evaluation, this assessment will not only look at the performance using data over the entire winter season (regardless of precipitation phase), it will also isolate the performance for snow.

## 2 Methods

Precipitation and ancillary meteorology data at 1-minute resolution for the 2015/2016 and 2016/2017 winter periods were obtained from the eight SPICE sites listed in Table 1. The data were quality controlled using the same techniques employed in SPICE (Nitu et al., 2018), which involved automated range and jump checks and supervised removal of remaining outliers. Next, the high-resolution precipitation data were subjected to the same Gaussian filter as the SPICE data used in K2017b to dampen high frequency noise; however, it was decided not to develop an event database such as the SEDS, but rather to use an alternate process to develop a consistent 30-minute precipitation time series to more closely reflect real-world precipitation datasets. This alternate process involves the use of a modified "Brute-Force" filter initially described in Pan et al. (2016), henceforth called the Neutral Aggregation Filter (NAF). NAF is described in more detail in Smith et al. (2019).

### 2.1 Neutral Aggregation Filter

The NAF algorithm removes noise in cumulative precipitation time series by iteratively balancing positive and negative noise and accumulating positive changes exceeding the noise by a user-defined threshold ($\Delta^*$, e.g. 0.05 or 0.2 mm, depending on the gauge precision) (Smith et al., 2019) such that the total accumulated positive increases in bucket weight after filtering are forced to equal the total end-of-season bucket weight. The algorithm removes random and systematic diurnal noise, but does not account for signal drift, for example that occurs due to evaporation of water within the gauge bucket. Signal drift can result in estimation errors, which can be mitigated using an iterative manual process, with the NAF output as a first guess. This process is called NAF Supervised (NAF-S) and lets the user select the beginning and the end points of segments within the time series where evaporation is occurring. The process then removes these segments so that they have no impact on the time series. Because there is some user subjectivity in selecting the beginning and end points of impacted segments, this process is completed by a single user employing pre-determined and consistent criteria. Although beyond the scope of the present work, testing NAF and NAF-S on both simulated and observed precipitation time series over an entire winter season, including both noise and evaporation drift, showed the technique to be effective with low error as compared to the control. The end product of the NAF-S filter is a clean, time consistent 1-minute accumulating time series with preserved data gaps (i.e. no gap filling) for each gauge configuration for each season.



## 2.2 Amalgamation and adjustment

To produce accumulation periods consistent with the K2017b validation, the 1-minute NAF-S precipitation accumulation time series were resampled to 30-minute accumulation amounts via differentiating the bucket weights between the start and end of the 30-minute period. The continuity of the time series was maintained, despite data gaps,

through the assumption that the gauge continues to accumulate precipitation despite logger or power outages. In the event of missing data, precipitation accumulation during outages was calculated based on the differential bucket weight between the start and end of the outage and recorded as an accumulation at the end of the outage. Although this preserves the total accumulation during the outage, information related to the timing of the events during the outage are not preserved, and the accumulation data need to be flagged. Protocols for adjusting the data for undercatch

are noted below.

The 1-minute wind speeds ($U_{10m}$ and $U_{gh}$ where available) and air temperatures (generally measured at 1.5 m) were averaged over the same 30-minute periods as the accumulated precipitation amounts. Site specific details on the ancillary measurements can be found in the SPICE site commissioning reports (referenced in Section 1). If more than 10 minutes of wind or temperature data were missing in any 30-minute period, the data were flagged as missing and

were not used in the adjustment procedure

The resultant time series for each test gauge at each site were adjusted separately using both Eq. 1 and Eq. 2. Each 30-minute accumulation was adjusted individually if the following conditions were met: 1) both the start and end bucket weights for the 30-minute period were not missing, such that the differential could be determined; and 2) no more than 10 of the 30-minute values of either wind speed or temperature were missing. Periods that did not meet these

criteria were preserved in the time series, but were flagged as being unadjusted and were not included in the validation.

For adjustments using Eq. 1, the pre-determination of the precipitation phase was not necessary, as the transfer function is continuous with temperature and not directly dependent on precipitation phase. For the purpose of adjusting precipitation using Eq. 2, phase is determined by air temperature using the phase regimes outlined in Section 1.1 with rain assuming a catch efficiency of 1 (and is therefore not adjusted). The same maximum wind speed thresholds for

adjustments were employed here as in K2017b, which were 7.2 m s$^{-1}$ and 9.0 m s$^{-1}$ at gauge height (generally 2 m above ground) and at 10 m height respectively. Wind speeds above these thresholds were set at the threshold value to avoid over adjustment and the increased uncertainty in the transfer functions above the wind speed threshold. The resulting data includes a sub-set of adjusted and unadjusted 30-minute precipitation amounts for each SUT gauge configuration at each site (adjusted using Eq. 1 and Eq. 2 and using either $U_{gh}$ or $U_{10m}$ or both where available), the

30-minute DFAR data, and the accumulated gap preserved time series for each with flags identifying the periods that were not adjusted.

The performance of adjustments were assessed using the relative total catch (RTC; defined as the total catch of the gauge under test as compared to the DFAR, expressed as a percentage), the root mean square error, Pearson correlation, , and the Nash-Sutcliffe Efficiency (NSE). The assessment considered overall performance for all

precipitation types combined, as well as for snow, alone. In the latter case, the assessment of snow adjustments using



Eq. 1 employed the same temperature threshold for snow as K2017b ($T_{air} < $ -2 °C). Intercomparison results are reported for both unshielded and single-Alter shielded sensors under test, which can be either a Geonor T-200B3 or OTT Pluvio[2]. Where multiple gauges of the same configuration are present at a site, these gauges are assessed individually and as a combined dataset.

For evaluation purposes, and where possible, the following circumstances will be assessed for both all precipitation types (includes rain, snow, and mixed) and snow only:  a) adjustments using Eq. 1 vs. Eq. 2, b) adjustments using gauge height vs. 10 m wind speeds, and c) adjusting single-Alter shielded vs. unshielded gauges. Based on site-by-site evaluations, some insight will be provided as to the performance of transfer functions in different environments, and under different climate characteristic conditions.

**3 Results**

**3.1 Time series**

The impact and the performance of transfer functions for adjusting precipitation can be examined by comparing the accumulation time series for unadjusted and adjusted data to the reference. Figures 3 and 4 show unadjusted, adjusted (Eq. 1 only), and reference (DFAR) time series of precipitation accumulation for the unshielded and single-Alter

shielded gauges at all sites for the 2015/2016 and 2016/2017 winter seasons, respectively. Where more than one gauge with the same shield configuration was present at a site, and where more than one wind speed height was available, results for only one gauge and wind speed height were selected for illustrative purposes.

Figures 3 and 4 show the relative impacts of wind on undercatch for each of the eight sites and the relative effectiveness of the transfer function adjustments on each SUT configuration (shielded and unshielded) for the two winter seasons

separately. Note that the season lengths vary by site and season (depending on both the actual length of the winter season and on data availability) and the scale of the vertical axis changes with site and season to show the relative scale of the bias and the adjustment. Gaps in the series represent missing data, with total accumulations during the gap obtained from the bucket weight change (in both the DFAR and the SUT); the gap accumulations were preserved but not adjusted.

The precipitation amounts vary by site and season, but the general trends in SUT undercatch as compared to the DFAR are consistent. Accordingly, the impact of the adjustment also appears to be consistent. Without considering precipitation phase partitioning, unadjusted precipitation (solid lines) relative to the DFAR was always lowest at the windy sites of XBK and HKL. Referring to unadjusted precipitation, unshielded gauges (blue lines) always catch less precipitation than the single-Alter shielded gauges (red lines) at all sites, and during both winters. The transfer function

(only temperature dependent Eq. 1 adjustments shown here) appears to be less effective at the windier sites (wind speeds during precipitation events are shown in Table 2), with a substantial undercatch remaining after adjustment. Further, precipitation is over-adjusted at some of the less windy sites (CAR, CCR, and SOD).



### 3.2 Relative total catch

The relative total catch (RTC) is an important performance metric for climate monitoring and analysis and reflects the capability of the transfer functions to correctly adjust seasonal and long term total precipitation. The RTC metrics for the single-Alter shielded gauges are shown in Table 2 and for the unshielded gauges in Table 3 for Eqs. 1 and 2,
combining the data from the two intercomparison seasons (CCR being the exception, since data is only available for 2016/2017). RTC is shown for both all precipitation phases and for snow, and for both wind measurement heights ($U_{10m}$ and $U_{gh}$), where available. If there are multiple gauges per site, the RTC is reported separately for each gauge and for the combined gauge dataset. Figure 5 summarizes the snow RTC in the tables for the combined results for both winter seasons. When more than one gauge exists, the $U_{gh}$ was used for the adjustment for all sites except FOR,
which only reported $U_{10m}$.

The RTC values in Tables 2 and 3 indicate that the unadjusted catch for snow is lower than the unadjusted catch for all precipitation types at most of the sites. The magnitude of the difference depends on the relative amounts of solid and liquid precipitation received during the season, as well as the wind speeds during snow. The biggest difference in the unadjusted RTC values between all precipitation types and snow occurred at sites where more rain occurred during
the intercomparison season, as the gauge catch for rainfall is naturally higher (Yang et al., 1998; Smith, 2008) and biases the total catch. At XBK, CAR, FOR, and HKL, removing rain and mixed precipitation from the statistics had a large impact on the RTC, both pre- and post-adjustment, and provides a more realistic metric for assessing how well the transfer functions are performing for the adjustment of snow measurements.

Although the sample size is smaller (fewer unshielded than single-Alter shielded gauges), the unadjusted catch of the
unshielded gauges (Table 3) was lower than the unadjusted catch of the single-Alter shielded gauges (Table 2). From the single-Alter shielded RTC in Table 2, focusing on snow, the differences between the Eq. 1 and Eq. 2 results were small, varying within 1 to 2 %. This can also be seen in the combined results in Fig. 5. The difference between Eq.1 and Eq. 2 was greater for the unshielded gauges (Table 3) with Eq. 1 performing better than Eq. 2 for snow at XBK (+12 %) and HKL (+10% using $U_{gh}$). Equation 2 tended to under-adjust the unshielded gauges more than Eq. 1 at
XBK and HKL.

At sites with both wind speed heights available for use in the adjustment (XBK, CAR, HKL, and MAR), the data shown in Table 2 for single-Alter shielded gauges suggest that using $U_{gh}$ reduces the extent of over-adjustment (CAR, MAR) or under-adjustment (XBK and HKL) relative to $U_{10m}$ (i.e. adjustments closer to 100%). This holds for the unshielded gauge adjustments in Table 3, with the exception of MAR, which shows a large under-adjustment using
$U_{gh}$ as compared to $U_{10m}$. This may suggest that wind speeds are biased low at MAR, which is consistent with comments made by Kochendorfer et al. 2017a stating that the ground height wind measurement may be shadowed in some directions. Although the sample size was small, there is reason to suggest from an RTC perspective that $U_{gh}$ outperforms $U_{10m}$ when used in Eq.1 and 2 for adjusting snow measurements. For this reason, where available, the $U_{gh}$ rather than $U_{10m}$ wind adjustments are shown in Fig. 5 and subsequent figures.





The differences in RTC for adjusted measurements from single-Alter shielded gauges versus those from unshielded gauges are mixed. At the windier HKL and XBK sites, Fig. 5 suggests that the adjusted RTC for the unshielded gauge is just as high as or higher than for the single-Alter shielded gauge (Eq. 2 at XBK being the exception). The over-adjustments at SOD and CCR are exaggerated for the unshielded gauge, but the unshielded adjustment is closer to

100% at CAR and WFJ. MAR is an outlier in Fig. 5, possibly due to the potential issue with $U_{gh}$, but the $U_{10m}$ adjustment of the unshielded gauge (Table 3, snow) has an RTC closer to 100% (105% and 106% for Eq. 1 and Eq. 2 respectively) than the single-Alter shielded gauge (117%).

Including the RTC for both wind speed heights but excluding the combined gauge statistics, the adjustment for snow using Eq. 1 increases the mean catch efficiency for the single-Alter shielded gauge from 61% to 88% and for the

unshielded gauge from 48% to 92%.

### 3.2 RMSE

The RMSE was used to estimate the uncertainty of unadjusted and adjusted precipitation measurements relative to the reference. However, the variability in the magnitude of the RMSE amongst the sites should be interpreted with caution, as a small RMSE at a site with low precipitation rates may be more significant than a higher RMSE at a site that has

higher precipitation rates. For that reason, the RMSE will mainly be used to assess the relative performance of Eq. 1 and Eq. 2 at each site, as well as the relative performance of the transfer functions for the single-Alter shielded and unshielded gauges.

Table 4 (single-Alter shielded gauges) and Table 5 (unshielded gauges) show the RMSE for each available SUT at each site, as well as the RMSE when multiple SUTs are combined. As with RTC, the metric is provided for both

transfer functions and using both wind speed heights, where available.

Comparing the RMSE values between Table 4 and Table 5, the RMSE values for both the adjusted and unadjusted unshielded gauges are higher than their single-Alter shielded counterparts. However, the RMSE differences between all precipitation phases and those for snow are inconsistent, with RMSE occasionally lower for snow than for all phases, and vice versa. For the single-Alter shielded and unshielded gauges, respectively, 46% and 36% of the RMSE

values are either lower or the same for snow as compared to all precipitation phases. The differences in adjusted RMSE with wind measurement height are also small. For single-Alter shielded gauges, CAR has a lower RMSE for $U_{gh}$, MAR has a higher RMSE for $U_{gh}$, and HKL and XBK show similar RMSE values for $U_{gh}$ and $U_{10m}$. For unshielded gauges, the RMSE for $U_{gh}$ is lower than that for $U_{10m}$ at XBK and CAR, and higher at HKL and MAR.

When considering each gauge at each site, the anticipated decrease in the RMSE following adjustment (whether for

all precipitation types or snow only) is not universal. This is illustrated in Fig. 6, which shows the RMSE results for combined SUT snow datasets before and after adjustment using $U_{10m}$ (where possible). For single-Alter shielded gauges, the RMSE increases with adjustment at CAR, MAR, SOD and WFJ (although the increase is small at all sites but WFJ). The decrease in RMSE is small at XBK and CCR. The differences between RMSE results using Eq. 1 and



Eq. 2 for single-Alter shielded gauges are insubstantial. These differences are larger for the unshielded gauges as shown in Fig. 6.

### 3.3 Pearson correlation

The Pearson correlation (r) assesses the strength of linear relationships between the SUT and the references before
and after adjustments using Eq. 1 and Eq. 2. Similar to previous metrics, the single-Alter shielded and unshielded r-values are shown separately in Tables 6 and 7 respectively, and are plotted for snow in Fig. 7. In theory, the adjustment using a transfer function should strengthen the linear relationship (increase r) between the adjusted and the reference measurements by removing the non-linearity associated with wind bias.

For single-Alter shielded gauges, unadjusted r-values for all precipitation types range from 0.83 at HKL to 0.96 at
CCR. For unshielded gauges, the unadjusted r-values for all precipitation types are only slightly lower than their shielded counterparts. The unadjusted r-values for single-Alter shielded gauges for snow are generally lower than for all precipitation types, especially at the windy sites of HKL and XBK. The unshielded, unadjusted values for snow follow similar trends as all precipitation types.

The results show that adjusting measurements for all precipitation types with either transfer function has little impact
on the r-values, with greater variability in r values observed for the unshielded gauges. The impact of adjustments on snow measurements are shown in Fig. 7. Generally, the r-values for the single-Alter shielded gauges are improved with adjustment, but the change is small ($< 0.07$). The largest improvements are observed for the HKL and XBK measurement datasets. With the unshielded adjustment, Unshielded gauges at most sites also show an improvement in r-value with adjustment, with the most significant increases observed for the HKL and MAR datasets.

For both all precipitation phases and snow only (Fig. 7), the differences between r-values following the application of Eq. 1 and Eq. 2 are negligible for the single-Alter shielded gauges. For the unshielded gauges, Eq. 2 results in higher r values than Eq. 1, but the differences are very small ($< 0.03$).

For sites with both wind speed measurement heights, the correlations appear to be independent from the measurement height. The only exception is for the unshielded adjustment of snow measurements at MAR, where correlations based
on transfer function application using $U_{gh}$ are significantly less than those using $U_{10m}$. This likely results from shadowing effects on the $U_{gh}$ data.

### 3.4 Nash-Sutcliffe Efficiency

The Nash-Sutcliffe Efficiency (NSE) coefficient is an alternative goodness-of-fit indicator for the agreement between the reference and the unadjusted and adjusted precipitation measurements. The measure has a higher sensitivity to
bias and outliers than the Pearson Correlation coefficient and this can have both advantages and disadvantages in this assessment. It is included here to complement the other metrics used in this analysis. The NSE coefficient can vary from $-\infty$ to 1, where 1 indicates a perfect fit between the reference and the SUT. The NSE for the single-Alter shielded





gauges and the unshielded gauges are shown in Tables 8 and 9 respectively and the values for snow are shown in Fig. 8.

Unadjusted NSE for single-Alter shielded gauges ranges from 0.55 to 0.90 for all precipitation and from 0.08 to 0.85 for snow. For the unshielded gauges, the NSE ranges from 0.36 to 0.83 for all precipitation and -0.55 to 0.75 for snow.

Similiar to the Pearson Correlation analysis, post-adjusted NSE for the single-Alter shielded gauges changes only a small amount with larger but inconsistent changes in the post-adjusted NSE for unshielded gauges. For unshielded gauges, some sites show a marked improvement in NSE after adjustment (HKL, MAR, and WFJ) while some sites show substantial decreases in NSE after adjustment (XBK and CAR). The unshielded NSE coefficients for XBK, both adjusted and unadjusted, are the only values less than 0 (and are clipped in Fig. 8).

From Fig. 8, the impact of the adjustment for single-Alter gauges using either Eq. 1 or Eq. 2 is relatively small. However, where the post-adjusted r-values for unshielded gauges are often similar to those for the single-Alter gauges, most sites clearly show a lower post-adjusted unshielded NSE as compared to their single-Alter counterparts. As for the relative performance of Eq. 1 vs. Eq. 2, NSE shows little difference for the single-Alter shielded gauges with perhaps a slight advantage to Eq. 2 for adjusting the unshielded gauges.  Finally, for sites with both wind speed heights,

the differences in the NSE metric are more pronounced than for the Pearson Correlation, but results remain mixed and varied by site.

**4 Discussion**

The current application of the universal transfer functions developed in SPICE to two winter season of precipitation data at eight locations produced variable results depending on site location. The discussion will focus on snow to avoid

the confounding influence of precipitation phases, in varying proportions, on the assessment results. Based on the relative total catch results, the transfer functions tend to under-adjust snow for single-Alter shielded gauges at the windy sites of XBK and HKL with mean 10 m wind speed during snow of 6.1 and 5.3 m s$^{-1}$, respectively (Table 2). The results from FOR were similar, despite relatively lower mean wind speeds of 4 m s$^{-1}$ at 10 m. The SOD, CCR, WFJ, and MAR sites were characterized by the lowest wind speeds, with mean values at 10 m smaller than 3.2 m s$^{-1}$.

For single-Alter shielded gauges at these sites, the RTC following adjustment varied from 105% at SOD and CAR to 113% at MAR and WFJ. The above trends are similar to the bias results of K2017b for all precipitation phases, but more pronounced due to the focus on snow.

The adjusted RTC results showed greater variability for unshielded gauges, performing better at some sites and poorer at others relative to the performance for the single-Alter shielded gauges. Unexpectedly, the increase in RTC after

adjustment of the unshielded gauges at the windy sites (XBK and HKL) was quite effective. However, upon examination of the RMSE results, the errors associated with adjusting the unshielded gauges were generally higher compared to those associated with the single-Alter shielded gauges. The NSE points to this as well. Since the unadjusted RTC of unshielded gauges are lower (especially at windy sites), the adjustment for unshielded measurements must be larger, so signal noise and other random measurement errors after adjustment are magnified.

This also applies to very windy sites, where the required magnitude of the adjustment is necessarily larger. This propagation of errors through adjustments is discussed in greater detail by Kochendorfer et al. (2018).

The evaluation of transfer function performance is complicated by observations for which the DFAR detects a measurable amount of precipitation, but the SUT does not. A gauge measurement of zero cannot be adjusted with a transfer function. This impacts the performance metrics and cannot be ignored in the context of real-world applications of transfer functions (e.g. adjusting precipitation measurements for use in forecast validation). The limited utility of transfer functions in this regard is due to the configuration's capability to catch snow and not because of the transfer function. To assess the relative impact of these types of events on the evaluations, descriptive statistics were calculated for events at XBK, HKL, and SOD.

For XBK, there were 498 30-minute snow events during which the DFAR measured an accumulation value greater than zero. The single-Alter shielded gauge did not report precipitation during 285 of those events, which accounted for 14% of the total DFAR accumulation over both measurement seasons, and had a mean wind speed of 6 m s$^{-1}$ (7.5 m s$^{-1}$) at gauge height (10 m). The number of events during which the unshielded gauge did not report precipitation was even higher: 376 events in total, accounting for 24% of the total DFAR precipitation, characterized by lower mean wind speeds (5.4 m s$^{-1}$ at gauge height and 6.8 m s$^{-1}$ at 10 m). For HKL, 860 of 1881 snow events were not reported by the single-Alter shielded gauge (8% of the total DFAR accumulation, and 966 of the 1881 events were not reported by the unshielded gauge (10% of the total DFAR accumulation). One can speculate that the influence of missed reports was more significant at XBK due to the drier nature of the site and of the falling snow, which made the snow more susceptible to deflection around the gauge inlet. At SOD, where wind speed has considerably less impact on gauge catch, 413 of 1656 events reported by DFAR were not reported by the single-Alter shielded gauge (about 6% of the total DFAR accumulation). The number of occurrences for the unshielded gauge at SOD was nearly identical to the single-Alter shielded gauge. At windy sites, when precipitation goes undetected by a non-reference gauge, there is a negative impact on the transfer function performance metrics, but it also means that the effectiveness of the transfer function is reduced when it is applied to operational observations using those same gauge configurations. Since more shielding (e.g. double-Alter) generally means a higher catch (Watson et al., 2008; Smith, 2009; Rasmussen et al., 2012; Kochendorfer et al., 2017b), more shielding would also reduce the number of unmeasured events. Even though some adjusted RTC values are closer to 100% for unshielded gauges (such as at HKL, CAR and WFJ), the RMSE was generally lower and the NSE consistently higher for single-Alter shielded gauges, and combined with a lower frequency of missed measurements, supports the use of more shielding for solid precipitation measurements.

In general, the application of transfer functions resulted in under-adjustment at the windier sites and over-adjustment at the less windy sites; however, there is no clear relationship between the mean wind speed at a site and transfer function performance. The general performance of the transfer functions for single-Alter shielded gauge measurements, from the perspective of RTC, likely also depends on other factors, such as crystal characteristics (Thériault et al., 2012) or aerodynamic peculiarities at the intercomparison sites affecting the representative wind speed measurements (as discussed in K2017b). Based on the present results, we found that the transfer function performance varied by site and the windy sites were under-adjusted while the less windy sites were over-adjusted, but



the magnitude of the adjustment error and the specific causes of error were difficult to determine. Although beyond the scope of this work, an alternative to universal transfer functions may be to develop site-specific transfer functions. Applicability to other sites with similar conditions could be assessed using a site classification process based on climate parameters and principle components analysis, such as that shown in Pierre et al. (2019).

The differences in performance between Eq. 1 and Eq. 2 for adjusting snow measurements from single-Alter shielded gauges were small: RTC generally varied by less than 2%, and RMSE, r and NSE were nearly identical. This was likely an artifact of the way that phase was determined in the methodology; even though $C_E$ is a function of air temperature in Eq. 1, the data for snow are a subset of the precipitation data based on the same phase discrimination used for Eq. 2. However, the metrics for all precipitation types were also similar, which is consistent with the results

in K2017b, and suggests that transfer function selection is essentially a matter of user preference. In that respect, the "simpler" Eq. 2 is less simple in that user is required to determine the phase based on temperature thresholds, while Eq. 1 requires no phase discrimination. The decision would appear to be more complicated for unshielded gauges, likely because of the increased uncertainty in both the measurement and the adjustment. Generally, the differences are still quite small but Eq. 2 shows a slight advantage with RMSE and r with a more obvious advantage in NSE. As with

the single-Alter shielded gauges, the decision likely should be based on personal preference. It would be interesting, however, to explore refining the coefficients for Eq. 2 using optical disdrometers or present weather sensors to identify dominant precipitation phase and employing such instruments when performing adjustments. It may also be worthwhile to assess the performance of Eq. 2 while using hydrometeor temperature approximation, as described in Harder & Pomeroy (2013), for phase discrimination.

Only four of eight test sites measured both $U_{10m}$ and $U_{gh}$, so it is difficult to draw conclusions regarding the influence of wind speed measurement height on the performance of the transfer functions. For those four sites, the RTC using $U_{gh}$ was closer to 100% for many of the adjustments, but RMSE and NSE values varied and r-values were nearly identical. The $U_{gh}$ is a direct measurement of the wind speed at gauge height, and does not rely on the potentially problematic assumption that wind speed at 10 m is representative of wind speed at gauge height. This assumption

relies on the estimation of surface roughness, which changes with vegetation cover, snowfall, and drifting. It also neglects the impact of increasing snow depth on the relationship between the gauge height wind speed and the 10 m height wind speed. However, depending on the distance between the SUT and the $U_{gh}$ measurement, the $U_{gh}$ measurement may also not be representative of the wind speed at the SUT due to interference from instruments, wind shields, and other obstructions between the wind sensor and the gauge. As noted in K2017b, discrepancies in the

various wind speed measurements (whether instrument, height, or exposure related) make it difficult to ascertain any advantage or disadvantage of using one wind speed height over the other. It is recommended to use the best wind speed data available at a given site for transfer function adjustment, but to be cognizant of the issues related to spatial representation of wind speed at the site.



## 5 Conclusions

The evaluation of the performance of WMO SPICE transfer functions using an independent, post-SPICE dataset showed that the performance varies by site and shield configuration and is considerably reduced when only assessing their performance for snow. Generally, the application of the transfer functions to measurements from sites with higher wind speeds resulted in an under-adjustment, while producing an over-adjustment for measurements from less windy sites. This trend was not universal, which indicates that the performance is also linked to local climatic conditions affecting snowfall characteristics. On average, the transfer functions resulted in an increase in the RTC of snow measurements from single-Alter shielded gauges (unshielded gauges) from 61% (48%) to 88% (92%), but also produced an under-adjustment as low as 54% and an over-adjustment as high as 123%. Although the RTC values imply improved transfer function performance when adjusting unshielded gauges relative to single-Alter shielded gauges, the higher RMSE and lower correlation and NSE for unshielded adjustments suggest otherwise. Further, the unshielded gauges were shown to completely miss a larger proportion of events and accumulated precipitation relative to the DFAR than the shielded gauges, raising the critical point that precipitation that is not recorded by the gauge configuration cannot be adjusted. The differences in performance observed for Eq. 1 and Eq. 2 were small enough that the choice of transfer function should largely depend on the availability of observed precipitation phase data as well as user preference. With only four sites collecting wind speed data at both 10 m and gauge height, it was difficult to determine if the wind speed measurement height significantly affected transfer function performance. RTC was generally closer to 100% when gauge height winds were used for the adjustment, but the RMSE, NSE and correlation results were mixed. Regardless, and perhaps more importantly, users must also carefully consider potential issues with obstructions and spatial representativeness when selecting a wind speed measurement.

Ultimately, eight DFAR intercomparison sites were insufficient to address the variability in performance of the SPICE transfer functions and more intercomparison sites with a DFAR are needed in various cold region climate regimes for more thorough assessments, a key recommendation from the WMO-SPICE project (Nitu et al., 2018) For the most part, and especially at locations that experience relatively high wind speeds during snowfall events, the application of the adjustment improved the usability of the observations. This study also suggests a high degree of uncertainty in applying these adjustments in networks that geographically span many different climate regimes, and additional work is required to assess and minimize that uncertainty.

## Data availability

The quality controlled 30-minute data set used in this publication will be made available via an online repository prior to the publication of this manuscript.

## Author contribution

C.S. is the lead author and completed the bulk of this analysis. A.R. was responsible for data management including coding, quality control and processing. J.K. provided advice and expertise on the use and assessment of the transfer functions. M.E. provided advice on analysis and manuscript development as well as contributing to the design and



implementation of the WMO-SPICE data quality control procedures used on these data. J.K., M.E., M.W., S.B., Y.R., T.L., and C.S. were core participants of the WMO-SPICE data team and were instrumental in the collection and provision of these data.

**Competing interests**

The authors declare that they have no conflict of interest.

**Acknowledgments**

The authors would like to thank Eva Mekis of Environment and Climate Change Canada for providing an internal review of this manuscript. We would like to acknowledge the organizations that collected and provided the data for this analysis: Environment and Climate Change Canada (Climate Research Division and the Meteorological Service

of Canada Observing Systems and Engineering) for XBK, CAR, and CCR; the National Center for Atmospheric Research and the National Oceanic and Atmospheric Administration for MAR; the Norwegian Meteorological Institute for HKL; MeteoSwiss for WFJ; the Finnish Meteorological Institute for SOD; and the Spanish National Meteorological Agency for FOR.

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





**Table 1: Site name (and country), abbreviation (Abbr.), latitude, elevation above sea level, and mean daily air temperature.**

| Site (Country) | Abbr. | Lat. | Elev. | Mean $T_{air}$* |
|---|---|---|---|---|
| Bratt's Lake (Canada) | XBK | 50.20° | 585 m | -2.1°C |
| CARE (Canada) | CAR | 44.23° | 251 m | -0.9 °C |
| Caribou Creek (Canada) | CCR | 53.94° | 519 m | -4.5 °C |
| Formigal (Spain) | FOR | 42.76° | 1800 m | -2.1 °C |
| Haukeliseter (Norway) | HKL | 59.81° | 991 m | -2.0 °C |
| Marshall (USA) | MAR | 39.59° | 1742 m | 4.0 °C |
| Sodankylä (Finland) | SOD | 67.37° | 179 m | -0.4 °C |
| Weissfluhjoch (Switzerland) | WFJ | 46.83° | 2537 m | -4.4 °C |

*daily average mean air temperature over the year from site climatology (Nitu et al., 2018)





**Table 2: Relative total catch (RTC) for the single-Alter shielded gauges (G for Geonor, P for Pluvio²) as compared to the DFAR reference at each site for the combined winters of 2015/216 and 2016/2017. Results are provided for available wind speed measurement heights ($U_{10m}$ and $U_{gh}$) and for Eq. 1 and Eq. 2. Metrics are separated by precipitation phase (all and snow only). Where both wind speed heights are available, the $U_{gh}$ data are shown in red and italicized.**

| Site | DFAR | Gauge | Wind Height | All precipitation phases | | | | Snow only | | | |
|---|---|---|---|---|---|---|---|---|---|---|---|
| | | | | Unadjusted RTC (%) | Eq. 1 RTC (%) | Eq. 2 RTC (%) | Mean Wind m s⁻¹ | Unadjusted RTC (%) | Eq. 1 RTC (%) | Eq. 2 RTC (%) | Mean Wind m s⁻¹ |
| XBK | G | G | 10 m | 63 | 79 | 77 | 6.4 | 40 | 69 | 67 | 6.1 |
| | | G | *gh* | 63 | *80* | *79* | 4.9 | 42 | *71* | *70* | 4.8 |
| CAR | P | G | 10 m | 86 | 107 | 105 | 4.2 | 75 | 112 | 112 | 4.2 |
| | | P | 10 m | 85 | 105 | 104 | 4.3 | 71 | 107 | 106 | 4.3 |
| | | C | 10 m | 86 | 106 | 105 | 4.2 | 73 | 109 | 109 | 4.2 |
| | | G | *gh* | 86 | *104* | *104* | 3.2 | 75 | *108* | *109* | 3.3 |
| | | P | *gh* | 85 | *102* | *102* | 3.2 | 71 | *102* | *103* | 3.3 |
| | | C | *gh* | 86 | *103* | *103* | 3.2 | 73 | *105* | *106* | 3.3 |
| CCR* | G | G | gh | 82 | 110 | 109 | 2.3 | 83 | 110 | 110 | 2.2 |
| FOR | P | P | 10 m | 72 | 82 | 82 | 3.1 | 46 | 61 | 61 | 4.0 |
| HKL | G | G | 10 m | 57 | 74 | 72 | 5.7 | 43 | 62 | 61 | 5.3 |
| | | G | 10 m | 58 | 76 | 75 | 5.7 | 44 | 65 | 64 | 5.2 |
| | | P | 10 m | 55 | 71 | 69 | 5.8 | 38 | 55 | 54 | 5.4 |
| | | C | 10 m | 57 | 74 | 72 | 5.7 | 42 | 61 | 60 | 5.3 |
| | | G | *gh* | 57 | *80* | *78* | 5.6 | 43 | *70* | *69* | 5.2 |
| | | G | *gh* | 58 | *82* | *80* | 5.6 | 44 | *73* | *72* | 5.1 |
| | | P | *gh* | 55 | *77* | *75* | 5.7 | 38 | *62* | *61* | 5.3 |
| | | C | *gh* | 57 | *80* | *78* | 5.6 | 42 | *69* | *67* | 5.2 |
| MAR | G | G | 10 m | 83 | 107 | 105 | 3.8 | 84 | 117 | 117 | 3.2 |
| | | G | *gh* | 85 | *101* | *101* | 2.2 | 86 | *113* | *113* | 2.0 |
| SOD | P | P | gh | 92 | 104 | 104 | 1.5 | 91 | 106 | 105 | 1.3 |
| WFJ | P | P | gh | 82 | 111 | 111 | 2.5 | 79 | 113 | 113 | 2.8 |

5    *2016/2017 winter only





**Table 3: Relative total catch (RTC) for the unshielded gauges (G for Geonor, P for Pluvio[2]) as compared to the DFAR reference at each site for the combined winters of 2015/216 and 2016/2017. Results are provided for available wind speed measurement heights (U[10m] and U[gh]) and for Eq. 1 and Eq. 2. Metrics are separated by precipitation phase (all and snow only). Where both wind speed heights are available, the U[gh] data are shown in red and italicized.**

| Site | DFAR | Gauge | Wind Height | All precipitation phases | | | | Snow only | | | |
|------|------|-------|-------------|--------------------------|--|--|--|-----------|--|--|--|
| | | | | Unadjusted RTC (%) | Eq. 1 RTC (%) | Eq. 2 RTC (%) | Mean Wind m s⁻¹ | Unadjusted RTC (%) | Eq. 1 RTC (%) | Eq. 2 RTC (%) | Mean Wind m s⁻¹ |
| XBK | G | G | 10 m | 53 | 76 | 72 | 6.4 | 23 | 70 | 58 | 6.1 |
| | | G | *gh* | 53 | *79* | 74 | 4.9 | 23 | *69* | *58* | 4.8 |
| CAR | P | G | 10 m | - | - | - | 4.2 | - | - | - | 4.2 |
| | | P | 10 m | 75 | 106 | 104 | 4.3 | 50 | 115 | 109 | 4.3 |
| | | C | 10 m | - | - | - | 4.2 | - | - | - | 4.2 |
| | | G | gh | - | - | - | 3.2 | - | - | - | 3.3 |
| | | P | *gh* | 75 | *102* | *101* | 3.2 | 48 | *98* | *96* | 3.3 |
| | | C | gh | - | - | - | 3.2 | - | - | - | 3.3 |
| CCR* | G | G | gh | 72 | 125 | 121 | 2.3 | 72 | 123 | 122 | 2.2 |
| FOR | P | P | 10 m | | - | - | 3.1 | - | - | - | 4.0 |
| HKL | G | G | 10 m | 47 | 71 | 68 | 5.7 | 32 | 63 | 61 | 5.3 |
| | | G | 10 m | - | - | - | 5.7 | - | - | - | 5.2 |
| | | P | 10 m | - | - | - | 5.8 | - | - | - | 5.4 |
| | | C | 10 m | - | - | - | 5.7 | - | - | - | 5.3 |
| | | G | *gh* | 46 | *85* | *77* | 5.6 | 32 | *83* | *73* | 5.2 |
| | | G | gh | - | - | - | 5.6 | - | - | - | 5.1 |
| | | P | gh | - | - | - | 5.7 | - | - | - | 5.3 |
| | | C | gh | - | - | - | 5.6 | - | - | - | 5.2 |
| MAR | G | G | 10 m | 60 | 89 | 86 | 3.8 | 58 | 105 | 106 | 3.2 |
| | | G | *gh* | 60 | *78* | *78* | 2.2 | 50 | *79* | *80* | 2.0 |
| SOD | P | P | gh | 83 | 104 | 105 | 1.5 | 83 | 111 | 111 | 1.3 |
| WFJ | P | P | gh | 59 | 100 | 97 | 2.5 | 53 | 101 | 97 | 2.8 |

5    *2016/2017 winter only





**Table 4: Root mean square error (RMSE) for the single-Alter shielded gauges (G for Geonor, P for Pluvio[2]) as compared to the DFAR reference at each site for the combined winters of 2015/216 and 2016/2017 for available wind speed measurement heights (U$_{10m}$ and U$_{gh}$) and for Eq. 1 and Eq. 2. Metrics are separated by precipitation phase (all and snow only). Where both wind speed heights are available, the U$_{gh}$ data are shown in red and italicized.**

| Site | DFAR | Gauge | Wind Height | All precipitation phases | | | Snow only | | |
|---|---|---|---|---|---|---|---|---|---|
| | | | | Unadjusted RMSE (mm) | Eq. 1 RMSE (mm) | Eq. 2 RMSE (mm) | Unadjusted RMSE (mm) | Eq. 1 RMSE (mm) | Eq. 2 RMSE (mm) |
| XBK | G | G | 10 m | 0.141 | 0.131 | 0.133 | 0.138 | 0.126 | 0.125 |
| | | G | *gh* | 0.141 | *0.130* | *0.131* | 0.138 | *0.126* | *0.125* |
| CAR | P | G | 10 m | 0.150 | 0.172 | 0.164 | 0.197 | 0.216 | 0.211 |
| | | P | 10 m | 0.149 | 0.184 | 0.172 | 0.149 | 0.166 | 0.161 |
| | | C | 10 m | 0.150 | 0.178 | 0.168 | 0.176 | 0.193 | 0.189 |
| | | G | *gh* | 0.150 | *0.157* | *0.157* | 0.197 | *0.203* | *0.203* |
| | | P | *gh* | 0.149 | *0.162* | *0.160* | 0.149 | *0.143* | *0.144* |
| | | C | *gh* | 0.150 | *0.160* | *0.158* | 0.176 | *0.177* | *0.177* |
| CCR* | G | G | gh | 0.103 | 0.100 | 0.097 | 0.095 | 0.095 | 0.096 |
| FOR | P | P | 10 m | 0.339 | 0.268 | 0.277 | 0.483 | 0.379 | 0.379 |
| HKL | G | G | 10 m | 0.293 | 0.244 | 0.248 | 0.348 | 0.289 | 0.289 |
| | | G | 10 m | 0.287 | 0.258 | 0.259 | 0.344 | 0.319 | 0.317 |
| | | P | 10 m | 0.301 | 0.265 | 0.270 | 0.358 | 0.307 | 0.306 |
| | | C | 10 m | 0.293 | 0.256 | 0.259 | 0.349 | 0.305 | 0.305 |
| | | G | *gh* | 0.293 | *0.245* | *0.251* | 0.348 | *0.288* | *0.290* |
| | | G | *gh* | 0.287 | *0.266* | *0.267* | 0.344 | *0.328* | *0.329* |
| | | P | *gh* | 0.301 | *0.270* | *0.276* | 0.358 | *0.305* | *0.308* |
| | | C | *gh* | 0.293 | *0.260* | *0.265* | 0.350 | *0.308* | *0.309* |
| MAR | G | G | 10 m | 0.285 | 0.235 | 0.242 | 0.132 | 0.125 | 0.128 |
| | | G | *gh* | 0.273 | *0.243* | *0.251* | 0.144 | *0.151* | *0.153* |
| SOD | P | P | gh | 0.070 | 0.072 | 0.073 | 0.072 | 0.076 | 0.075 |
| WFJ | P | P | gh | 0.169 | 0.216 | 0.213 | 0.180 | 0.231 | 0.228 |

5    *2016/2017 winter only





**Table 5: Root mean square error (RMSE) for the unshielded gauges (G for Geonor, P for Pluvio[2]) as compared to the DFAR reference at each site for the combined winters of 2015/216 and 2016/2017 for available wind speed measurement heights (U$_{10m}$ and U$_{gh}$) and for Eq. 1 and Eq. 2. Metrics are separated by precipitation phase (all and snow only). Where both wind speed heights are available, the U$_{gh}$ data are shown in red and italicized.**

| Site | DFAR | Gauge | Wind Height | All precipitation phases | | | Snow only | | |
|---|---|---|---|---|---|---|---|---|---|
| | | | | Unadjusted RMSE (mm) | Eq. 1 RMSE (mm) | Eq. 2 RMSE (mm) | Unadjusted RMSE (mm) | Eq. 1 RMSE (mm) | Eq. 2 RMSE (mm) |
| XBK | G | G | 10 m | 0.172 | 0.205 | 0.185 | 0.180 | 0.245 | 0.202 |
| | | G | *gh* | 0.172 | *0.204* | *0.183* | 0.182 | *0.236* | *0.198* |
| CAR | P | G | 10 m | - | - | - | - | - | - |
| | | P | 10 m | 0.202 | 0.325 | 0.270 | 0.241 | 0.352 | 0.302 |
| | | C | 10 m | - | - | - | - | - | - |
| | | G | gh | - | - | - | - | - | - |
| | | P | *gh* | 0.202 | *0.277* | *0.249* | 0.241 | *0.301* | *0.271* |
| | | C | gh | - | - | - | - | - | - |
| CCR* | G | G | gh | 0.131 | 0.165 | 0.147 | 0.122 | 0.148 | 0.143 |
| FOR | P | P | 10 m | - | - | - | - | - | - |
| HKL | G | G | 10 m | 0.335 | 0.252 | 0.270 | 0.382 | 0.283 | 0.287 |
| | | G | 10 m | - | - | - | - | - | - |
| | | P | 10 m | - | - | - | - | - | - |
| | | C | 10 m | - | - | - | - | - | - |
| | | G | *gh* | 0.335 | *0.293* | *0.293* | 0.382 | *0.344* | *0.320* |
| | | G | gh | - | - | - | - | - | - |
| | | P | gh | - | - | - | - | - | - |
| | | C | gh | - | - | - | - | - | - |
| MAR | G | G | 10 m | 0.447 | 0.347 | 0.373 | 0.216 | 0.135 | 0.142 |
| | | G | *gh* | 0.411 | *0.345* | *0.357* | 0.284 | *0.240* | *0.242* |
| SOD | P | P | gh | 0.084 | 0.080 | 0.083 | 0.084 | 0.087 | 0.087 |
| WFJ | P | P | gh | 0.310 | 0.274 | 0.242 | 0.336 | 0.296 | 0.259 |

5    *2016/2017 winter only



**Table 6: Pearson correlation (r) for the single-Alter shielded gauges (G for Geonor, P for Pluvio[2]) as compared to the DFAR reference at each site for the combined winters of 2015/216 and 2016/2017 for available wind speed measurement heights ($U_{10m}$ and $U_{gh}$) and for Eq. 1 and Eq. 2. Metrics are separated by precipitation phase (all and snow only). Where both wind speed heights are available, the $U_{gh}$ data are shown in red and italicized.**

| Site | DFAR | Gauge | Wind Height | All precipitation phases | | | Snow only | | |
|---|---|---|---|---|---|---|---|---|---|
| | | | | Unadjusted r | Eq. 1 r | Eq. 2 r | Unadjusted r | Eq. 1 r | Eq. 2 r |
| XBK | G | G | 10 m | 0.89 | 0.89 | 0.89 | 0.67 | 0.71 | 0.71 |
| | | G | *gh* | 0.89 | *0.89* | *0.89* | 0.67 | *0.70* | *0.71* |
| CAR | P | G | 10 m | 0.96 | 0.95 | 0.95 | 0.83 | 0.86 | 0.86 |
| | | P | 10 m | 0.96 | 0.95 | 0.96 | 0.94 | 0.94 | 0.94 |
| | | C | 10 m | 0.96 | 0.95 | 0.96 | 0.88 | 0.90 | 0.90 |
| | | G | *gh* | 0.96 | *0.96* | *0.95* | 0.83 | *0.86* | *0.86* |
| | | P | *gh* | 0.96 | *0.96* | *0.96* | 0.94 | *0.95* | *0.95* |
| | | C | *gh* | 0.96 | *0.96* | *0.96* | 0.88 | *0.90* | *0.90* |
| CCR* | G | G | gh | 0.94 | 0.95 | 0.94 | 0.94 | 0.94 | 0.95 |
| FOR | P | P | 10 m | 0.92 | 0.94 | 0.94 | 0.86 | 0.89 | 0.89 |
| HKL | G | G | 10 m | 0.83 | 0.87 | 0.87 | 0.71 | 0.78 | 0.78 |
| | | G | 10 m | 0.83 | 0.85 | 0.85 | 0.70 | 0.72 | 0.72 |
| | | P | 10 m | 0.84 | 0.87 | 0.86 | 0.73 | 0.78 | 0.78 |
| | | C | 10 m | 0.83 | 0.86 | 0.86 | 0.71 | 0.76 | 0.76 |
| | | G | *gh* | 0.83 | *0.87* | *0.86* | 0.71 | *0.78* | *0.77* |
| | | G | *gh* | 0.83 | *0.85* | *0.85* | 0.70 | *0.72* | *0.72* |
| | | P | *gh* | 0.84 | *0.87* | *0.86* | 0.72 | *0.78* | *0.78* |
| | | C | *gh* | 0.83 | *0.86* | *0.86* | 0.71 | *0.76* | *0.75* |
| MAR | G | G | 10 m | 0.87 | 0.91 | 0.90 | 0.92 | 0.94 | 0.94 |
| | | G | *gh* | 0.86 | *0.89* | *0.88* | 0.90 | *0.91* | *0.91* |
| SOD | P | P | gh | 0.92 | 0.92 | 0.92 | 0.90 | 0.90 | 0.90 |
| WFJ | P | P | gh | 0.95 | 0.94 | 0.94 | 0.95 | 0.94 | 0.94 |

5    *2016/2017 winter only





**Table 7: Pearson correlation (r) for the unshielded gauges (G for Geonor, P for Pluvio[2]) as compared to the DFAR reference at each site for the combined winters of 2015/216 and 2016/2017 for available wind speed measurement heights ($U_{10m}$ and $U_{gh}$) and for Eq. 1 and Eq. 2. Metrics are separated by precipitation phase (all and snow only). Where both wind speed heights are available, the $U_{gh}$ data are shown in red and italicized.**

| Site | DFAR | Gauge | Wind Height | All precipitation phases | | | Snow only | | |
|---|---|---|---|---|---|---|---|---|---|
| | | | | Unadjusted r | Eq. 1 r | Eq. 2 r | Unadjusted r | Eq. 1 r | Eq. 2 r |
| XBK | G | G | 10 m | 0.84 | 0.75 | 0.79 | 0.30 | 0.28 | 0.32 |
| | | G | *gh* | 0.84 | *0.75* | *0.79* | 0.30 | *0.30* | *0.33* |
| CAR | P | G | 10 m | - | - | - | - | - | - |
| | | P | 10 m | 0.94 | 0.88 | 0.91 | 0.86 | 0.83 | 0.85 |
| | | C | 10 m | - | - | - | - | - | - |
| | | G | gh | - | - | - | - | - | - |
| | | P | *gh* | 0.94 | *0.90* | *0.92* | 0.86 | *0.84* | *0.86* |
| | | C | gh | - | - | - | - | - | - |
| CCR* | G | G | gh | 0.92 | 0.93 | 0.94 | 0.91 | 0.92 | 0.93 |
| FOR | P | P | 10 m | - | - | - | - | - | - |
| HKL | G | G | 10 m | 0.79 | 0.87 | 0.84 | 0.69 | 0.80 | 0.80 |
| | | G | 10 m | - | - | - | - | - | - |
| | | P | 10 m | - | - | - | - | - | - |
| | | C | 10 m | - | - | - | - | - | - |
| | | G | *gh* | 0.79 | *0.84* | *0.82* | 0.69 | *0.78* | *0.77* |
| | | G | gh | - | - | - | - | - | - |
| | | P | gh | - | - | - | - | - | - |
| | | C | gh | - | - | - | - | - | - |
| MAR | G | G | 10 m | 0.70 | 0.79 | 0.76 | 0.85 | 0.93 | 0.92 |
| | | G | *gh* | 0.72 | *0.79* | *0.77* | 0.65 | *0.71* | *0.70* |
| SOD | P | P | gh | 0.89 | 0.90 | 0.90 | 0.88 | 0.88 | 0.88 |
| WFJ | P | P | gh | 0.88 | 0.89 | 0.90 | 0.87 | 0.87 | 0.89 |

5    *2016/2017 winter only





**Table 8: Nash-Sutcliffe Efficiency (NSE) index for the single-Alter shielded gauges (G for Geonor, P for Pluvio²) as compared to the DFAR reference at each site for the combined winters of 2015/216 and 2016/2017 for available wind speed measurement heights (U$_{10m}$ and U$_{gh}$) and for Eq. 1 and Eq. 2. Metrics are separated by precipitation phase (all and snow only). Where both wind speed heights are available, the U$_{gh}$ data are shown in red and italicized.**

| Site | DFAR | Gauge | Wind Height | All precipitation phases | | | Snow only | | |
|------|------|-------|-------------|-----------|-------|-------|-----------|-------|-------|
| | | | | Unadjusted NSE | Eq. 1 NSE | Eq. 2 NSE | Unadjusted NSE | Eq. 1 NSE | Eq. 2 NSE |
| XBK | G | G | 10 m | 0.76 | 0.81 | 0.80 | 0.08 | 0.24 | 0.25 |
| | | G | *gh* | 0.74 | *0.79* | *0.78* | 0.09 | *0.25* | *0.26* |
| CAR | P | G | 10 m | 0.90 | 0.87 | 0.88 | 0.65 | 0.58 | 0.60 |
| | | P | 10 m | 0.91 | 0.85 | 0.87 | 0.80 | 0.76 | 0.77 |
| | | C | 10 m | 0.90 | 0.86 | 0.88 | 0.72 | 0.67 | 0.68 |
| | | G | *gh* | 0.90 | *0.89* | *0.89* | 0.65 | *0.63* | *0.63* |
| | | P | *gh* | 0.91 | *0.89* | *0.89* | 0.80 | *0.82* | *0.82* |
| | | C | *gh* | 0.90 | *0.89* | *0.89* | 0.72 | *0.72* | *0.72* |
| CCR* | G | G | gh | 0.85 | 0.86 | 0.87 | 0.85 | 0.85 | 0.85 |
| FOR | P | P | 10 m | 0.78 | 0.86 | 0.85 | 0.42 | 0.64 | 0.64 |
| HKL | G | G | 10 m | 0.56 | 0.69 | 0.68 | 0.26 | 0.49 | 0.49 |
| | | G | 10 m | 0.57 | 0.65 | 0.65 | 0.27 | 0.37 | 0.38 |
| | | P | 10 m | 0.55 | 0.65 | 0.64 | 0.23 | 0.44 | 0.44 |
| | | C | 10 m | 0.56 | 0.67 | 0.66 | 0.26 | 0.43 | 0.43 |
| | | G | *gh* | 0.56 | *0.69* | *0.67* | 0.26 | *0.49* | *0.49* |
| | | G | *gh* | 0.57 | *0.63* | *0.63* | 0.27 | *0.33* | *0.33* |
| | | P | *gh* | 0.55 | *0.64* | *0.62* | 0.23 | *0.44* | *0.43* |
| | | C | *gh* | 0.56 | *0.65* | *0.64* | 0.26 | *0.42* | *0.42* |
| MAR | G | G | 10 m | 0.73 | 0.82 | 0.81 | 0.80 | 0.82 | 0.81 |
| | | G | *gh* | 0.73 | *0.79* | *0.77* | 0.78 | *0.76* | *0.75* |
| SOD | P | P | gh | 0.84 | 0.83 | 0.83 | 0.80 | 0.78 | 0.79 |
| WFJ | P | P | gh | 0.87 | 0.80 | 0.80 | 0.85 | 0.76 | 0.76 |

5 *2016/2017 winter only





**Table 9: Nash-Sutcliffe Efficiency (NSE) index for the unshielded gauges (G for Geonor, P for Pluvio[2]) as compared to the DFAR reference at each site for the combined winters of 2015/216 and 2016/2017 for available wind speed measurement heights ($U_{10m}$ and $U_{gh}$) and for Eq. 1 and Eq. 2. Metrics are separated by precipitation phase (all and snow only). Where both wind speed heights are available, the $U_{gh}$ data are shown in red and italicized.**

| Site | DFAR | Gauge | Wind Height | All precipitation phases | | | Snow only | | |
|---|---|---|---|---|---|---|---|---|---|
| | | | | Unadjusted NSE | Eq. 1 NSE | Eq. 2 NSE | Unadjusted NSE | Eq. 1 NSE | Eq. 2 NSE |
| XBK | G | G | 10 m | 0.62 | 0.41 | 0.54 | -0.55 | -1.88 | -0.97 |
| | | G | *gh* | 0.58 | *0.38* | *0.51* | 0.30 | *-1.66* | *-0.87* |
| CAR | P | G | 10 m | - | - | - | - | - | - |
| | | P | 10 m | 0.83 | 0.55 | 0.69 | 0.49 | -0.83 | 0.20 |
| | | C | 10 m | - | - | - | - | - | - |
| | | G | gh | - | - | - | - | - | - |
| | | P | *gh* | 0.83 | *0.67* | *0.74* | 0.49 | *0.21* | *0.36* |
| | | C | gh | - | - | - | - | - | - |
| CCR* | G | G | gh | 0.76 | 0.61 | 0.69 | 0.75 | 0.63 | 0.65 |
| FOR | P | P | 10 m | - | - | - | - | - | - |
| HKL | G | G | 10 m | 0.42 | 0.67 | 0.62 | 0.11 | 0.51 | 0.50 |
| | | G | 10 m | - | - | - | - | - | - |
| | | P | 10 m | - | - | - | - | - | - |
| | | C | 10 m | - | - | - | - | - | - |
| | | G | *gh* | 0.42 | *0.55* | *0.55* | 0.11 | *0.28* | *0.37* |
| | | G | gh | - | - | - | - | - | - |
| | | P | gh | - | - | - | - | - | - |
| | | C | gh | - | - | - | - | - | - |
| MAR | G | G | 10 m | 0.36 | 0.62 | 0.56 | 0.45 | 0.79 | 0.76 |
| | | G | *gh* | 0.41 | *0.59* | *0.56* | 0.15 | *0.39* | *0.38* |
| SOD | P | P | gh | 0.78 | 0.79 | 0.78 | 0.74 | 0.72 | 0.72 |
| WFJ | P | P | gh | 0.58 | 0.67 | 0.74 | 0.48 | 0.60 | 0.70 |

5    *2016/2017 winter only





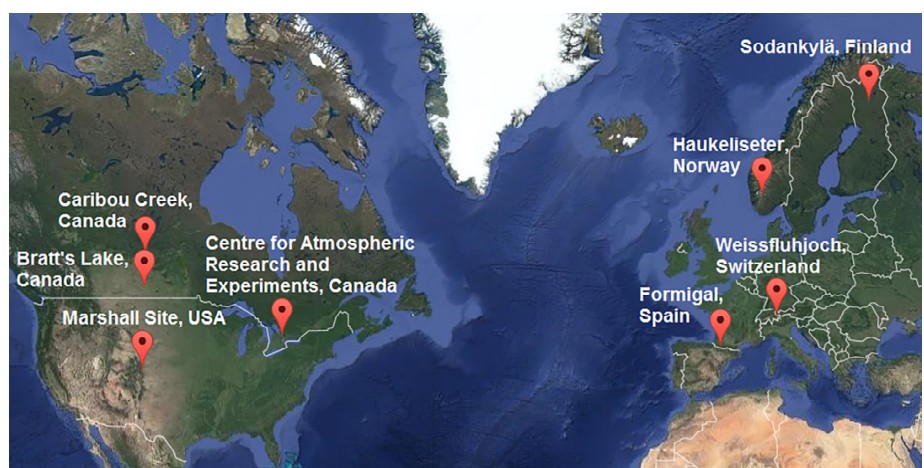

**Figure 1: The SPICE intercomparison sites used in the development and evaluation of the SPICE transfer functions (base map obtained from Google earth; Data SIO, U.S. Navy, GEBCO ©2018 Google; Image Landsat/Copernicus).**

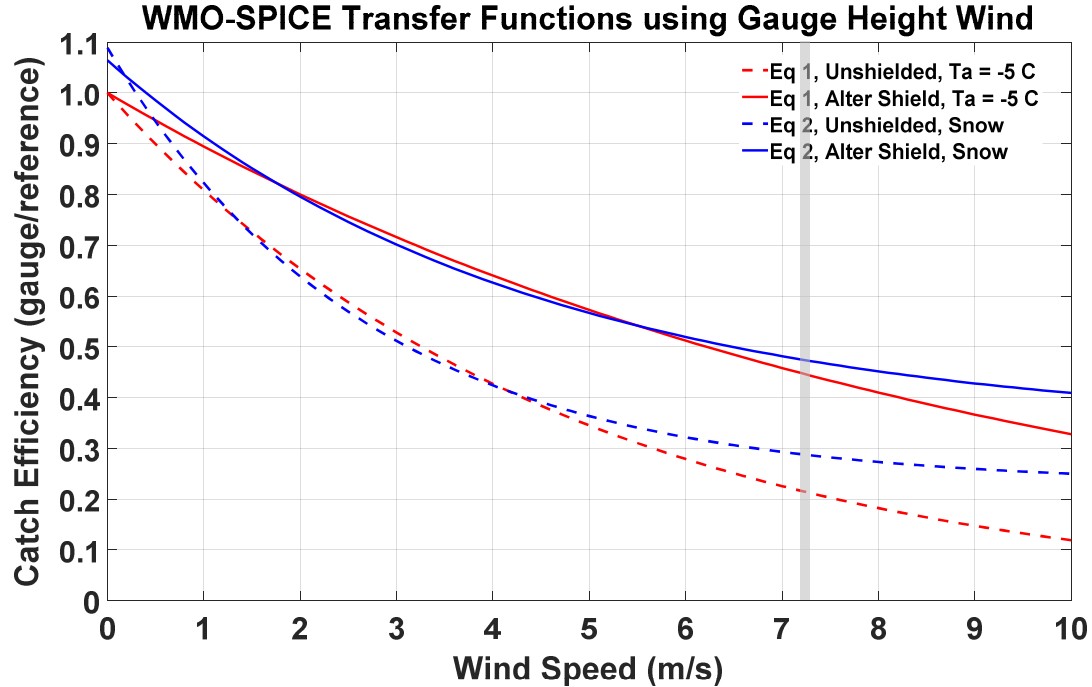

**Figure 2. WMO-SPICE transfer functions from K2017b, Eq. 1 (red), Eq. 2 (blue) for unshielded (dashed), and single-Alter shielded (solid) gauges. Eq. 1 is plotted using an air temperature of -5° C and Eq. 2 is plotted for snow. Both transfer functions are plotted for $U_{gh}$ with the maximum wind speed threshold shown at 7.2 m s$^{-1}$.**





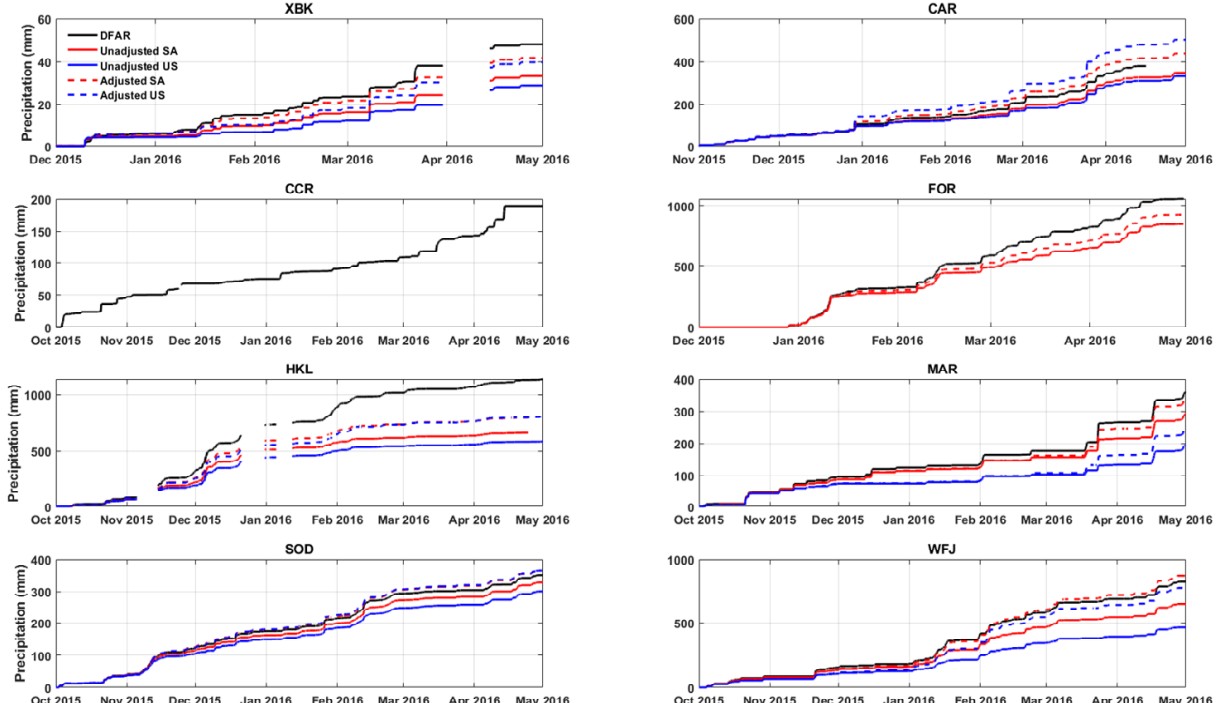

**Figure 3: Time series of unadjusted (solid) and adjusted (dashed) precipitation time series for single-Alter shielded (red) and unshielded (blue) gauges as compared to the DFAR (black) during the 2015/2016 winter season at the eight SPICE sites.**



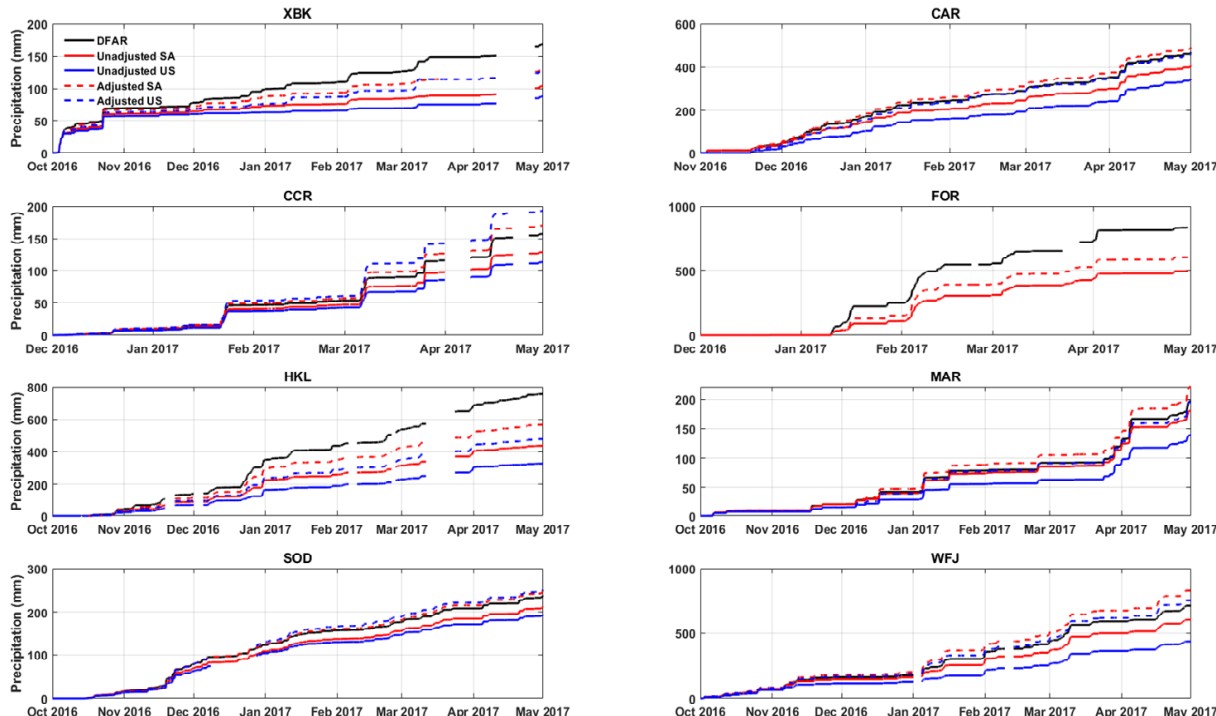

**Figure 4: Time series of unadjusted (solid) and adjusted (dashed) precipitation time series for single- Alter shielded (red) and unshielded (blue) gauges as compared to the DFAR (black) during the 2016/2017 winter season at the eight SPICE sites.**



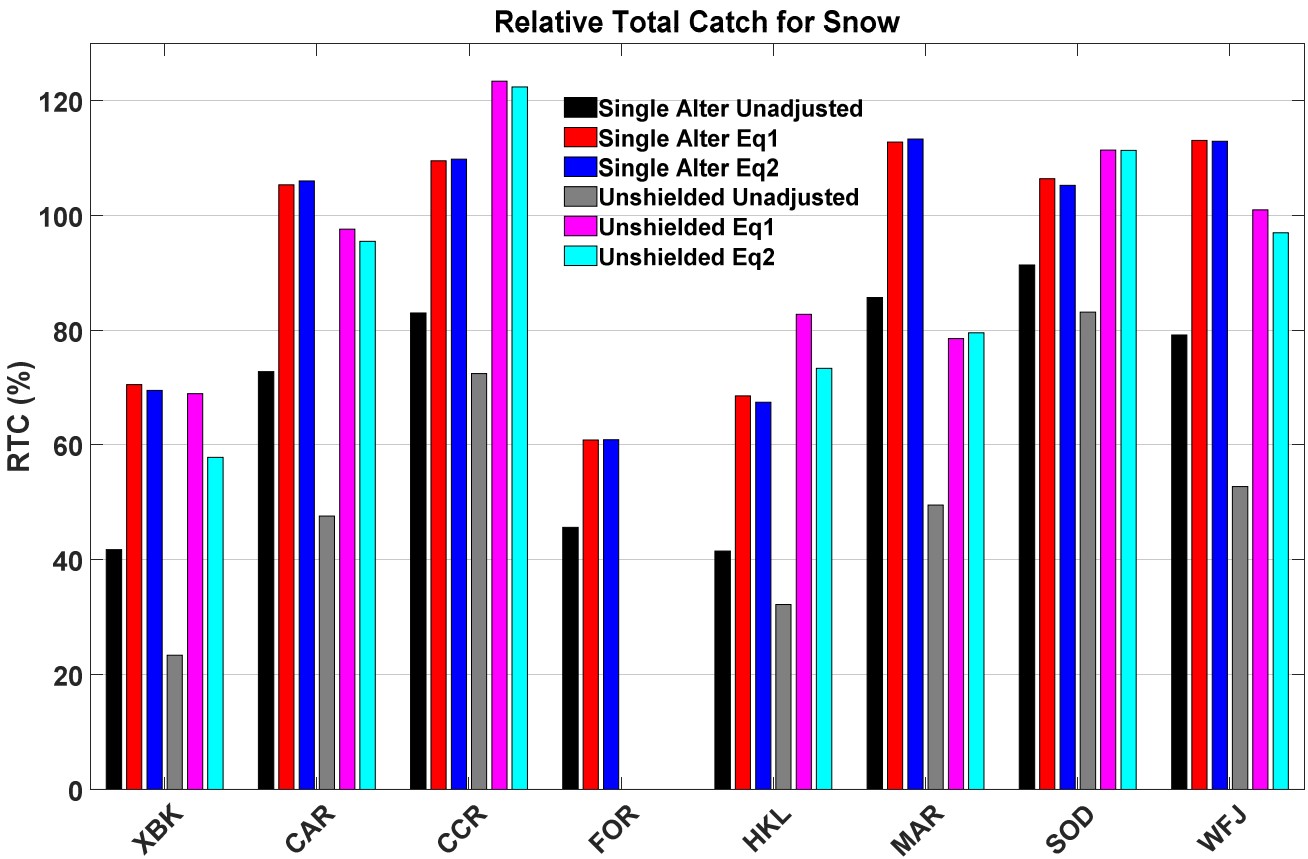

**Figure 5: Relative total catch of snow (as compared to the DFAR) for single single-Alter and unshielded gauges for both Equations 1 and 2 at each of the eight SPICE sites, combining 2015/2016 and 2016/2017 winter seasons.**



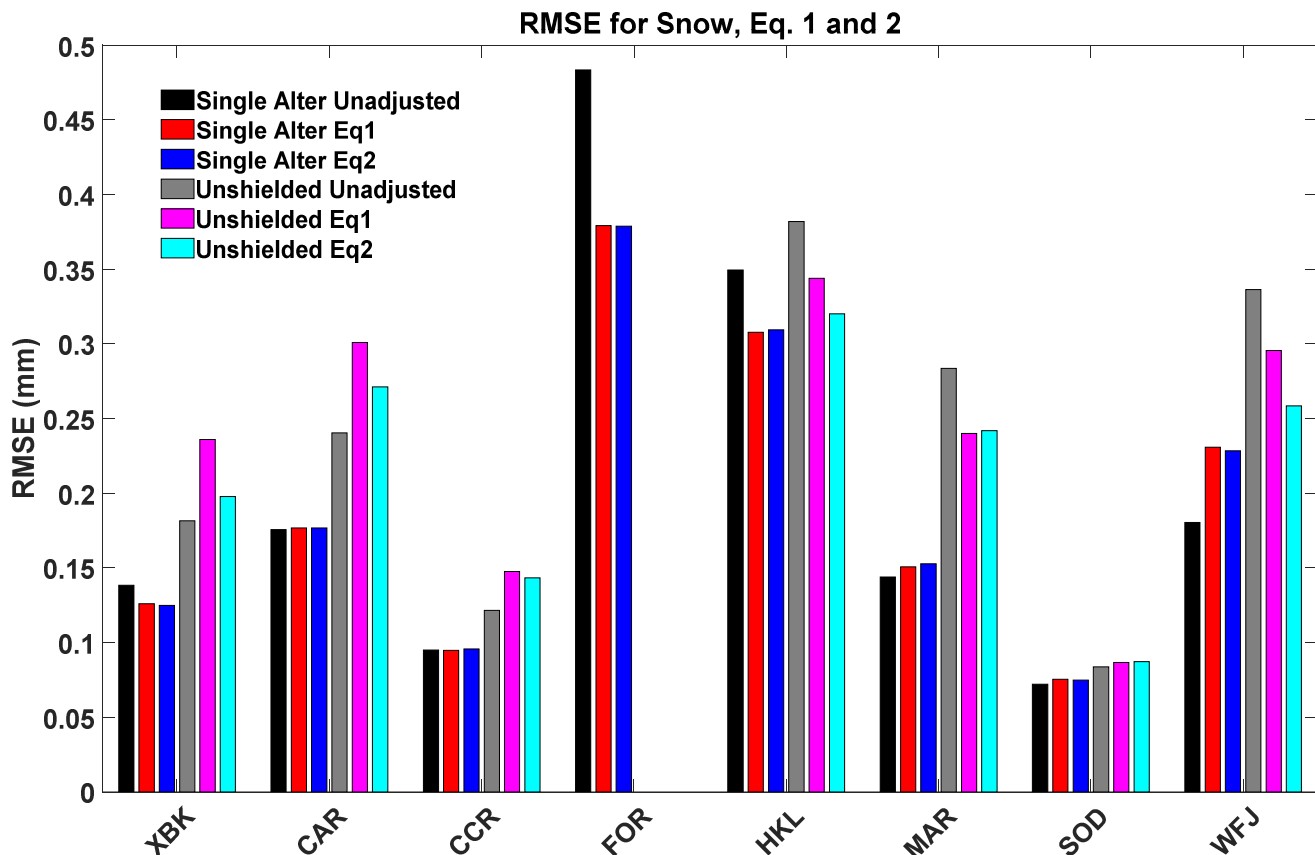

**Figure 6: RMSE for snow for single-Alter and unshielded gauges for unadjusted measurements and for adjustments using Equations 1 and 2 at each of the eight SPICE sites, combining 2015/2016 and 2016/2017 winter seasons.**



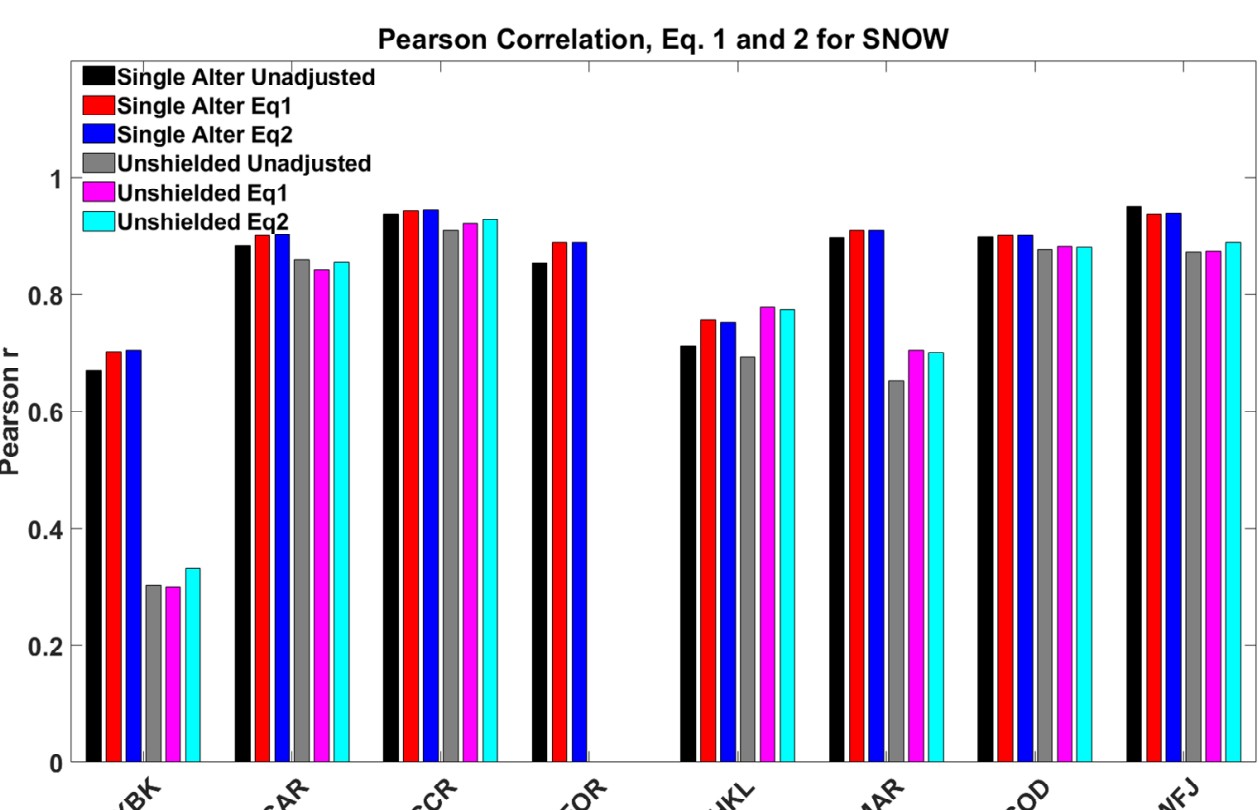

**Figure 7: Pearson r for single single-Alter and unshielded gauges for unadjusted measurements and for adjustments using Equations 1 and 2 at each of the eight SPICE sites, combining 2015/2016 and 2016/2017 winter seasons.**





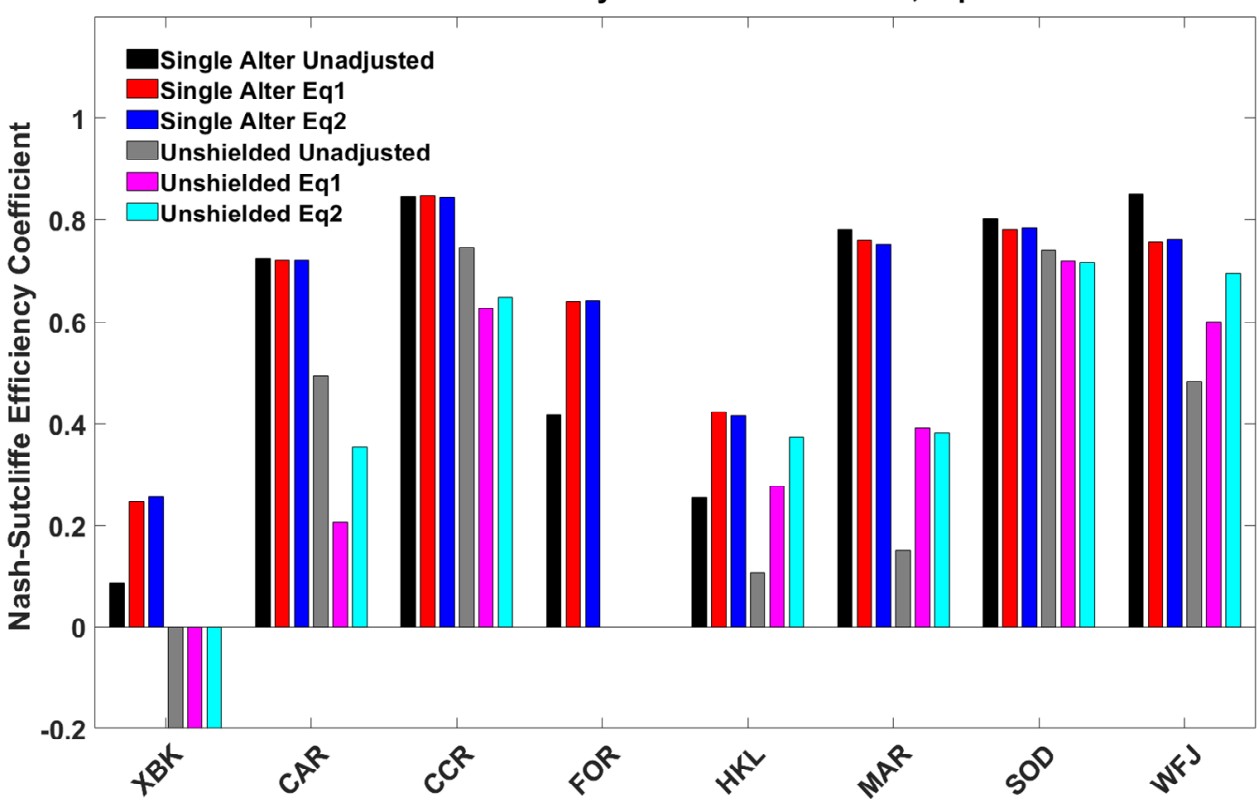

**Figure 8: Nash-Sutcliffe Efficiency Coefficient for single single-Alter and unshielded gauges for unadjusted measurements and for adjustments using Equations 1 and 2 at each of the eight SPICE sites, combining 2015/2016 and 2016/2017 winter seasons.**

