# Peer review of "Evaluation of the WMO-SPICE transfer functions for adjusting the wind bias in solid precipitation measurements"

_Hydrology and Earth System Sciences, 2019_

## Referee Comment (RC1) · Anonymous Referee #1 · 25 Oct 2019

The main purpose of this paper is to evaluate the performance of transfer functions, defined with 2 years of data, using the following 2 years of records in 8 locations that contain a great set of instruments to measure precipitation. The manuscript is well written and structured, and shows a critical analysis of the transfer functions in these 8 locations.

Minor comments:

P2, L15: In this paragraph, a short comment of how these stations are maintained after snowfall, or when they might get ice around, would be beneficial.

P2, L28: What is a Precipitation detector?

[Figure]

P3, L6: A reference after "demonstrating that the unshielded catch from both SUT types were very similar." would help in this sentence.

P4, L9-11: I think that a more detailed explanation of this sentence would help to complete the idea. For example, answering what makes these evaluations "marginally independent"?

P4, L17: I would include this sentence as the main result in the abstract.

P4, L29: What do you mean with "more natural circumstances"?

P4, L36: This has to be consistent with P4, L9-11.

P5, L3: I think the word "continued" should be in present tense.

P5, L7: I suggest completing the end of the sentence as: "...it will also isolate the performance to adjust snow measurements"

P9, L15: I have a question here, why not using a standardized RMSE value, as it highly depends on the precipitation rates?

P10, L1: Could you please add a number or a percentage to quantify the adjective "insubstantial"?

P9, L11-15; P10, L5 L7-8; P10, L28-32: these sentences describe the statistical parameters used to analyze the data. I suggest moving them to the methodology section.

P12, L10-11, The number of events that are not captured by the single-Alter shielded in XBK is over 50%, do you have any comments to propose a better monitoring system in this location? Or how we could possibly integrate these cases to the adjusted precipitation?

Table 1; Please, specify the meaning of "gh" in the table caption, or as a footnote.

Figures 3 and 4, I suggest increasing the panel size and/or font size.

---

## Referee Comment (RC2) · Anonymous Referee #2 · 6 Nov 2019

The focus of this manuscript is to test the transfer functions derived from WMO-SPICE on an independent set of data. The authors utilized data collected over two years from unshielded gauges, single Alter-shielded gauges and DFAR-shielded gauges. Overall, the manuscript is well-written and well-structured and the analysis is well explained and is an excellent follow-on to the initial work done by the SPICE group.

Major comments:

In the abstract and section 1.1, the authors discuss using liquid, mixed phase and solid precipitation. A possible useful addition to Table 1 would be statistics on the percentage of data that was liquid or mixed phase precipitation and the percentage of the data that

was solid precipitation for each site. This could potentially help explain some of the results you see in the other tables when comparing all precipitation phases to snow only.

The authors show analysis using the wind data collected from 10m heights and gauge height and discuss many issues related to measuring the wind speeds. In the discussion starting on page 13, line 20, as well as the first part of the conclusions section, no mention is made of the possible impacts of wind variations over the 30-minute averaging period. For the windier sites used in the study, how variable were the winds over the 30 minutes and what role might that have played in some of the results? I agree with the discussions and results the authors have expressed regarding some of the other wind-related issues but this should also be addressed.

Regarding the GEONOR gauges used in the study, the authors state they were GEONOR T-200B3 gauges. I believe this implies a three-wire GEONOR gauge. If that is the case, how were the data from the three wires used? Were they averaged? This should be discussed and/or clarified in the introduction or methods sections.

With regards to Fig 1, equation 2 shows a collection efficiency of 1.1 at a wind speed of 0, but I don't see a clear explanation for this. There is some discussion on P4, L1 that this might be indicative of rain at air temps > 2C, but equation 2 does not use temperature so I'm a bit confused as to why temperature is mentioned there and why eqn 2 would result in a catch efficiency > 1 at 0 m/s. Some clarification and additional explanation would be useful here.

In section 3.2 and 3.3, there is mention of potential shadowing of the wind sensor at the MAR site. Was anyone at the site contacted and asked for pictures of the sensor at the site to confirm this (assuming none of the authors have visited or work at this site)? This would seem to be an easy thing to do without supposition. Could this also be a calibration issue with the wind sensor at that site?

Minor comments:

Throughout the manuscript: There is inconsistent use of the hyphen in the phrase "single Alter shielded". In some places, there are two hyphens, in others, just one. I believe it should be "single Alter-shielded".

P1, L17 – Intercomparison is hyphenated here but nowhere else. The official SPICE title appears to not have the hyphen.

P1, L20 – The term "windshield" is typically used to describe the front glass on a car. In the case of precipitation measurement, I believe "wind shield" is more commonly used.

P1, L25-27, The sentence on these lines is somewhat awkwardly written.

P1, L29 – What is meant by gauge configuration?

P5, L24 – This sentence reads a bit awkwardly and may just be missing a comma. Perhaps revise into two sentences.

P6, paragraph 2 – How often were the temp/wind speed data missing?

P6, L34 – there is an extra comma on this line

P6, L35 – there shouldn't be a comma after "snow".

P7, L5 – The term "both" is not necessary in this sentence.

P8, L6 – The term "both" is not necessary here either.

P11, L18 – season should be plural.

P11, L20 – I think you meant to say compounding instead of confounding?

P12, L16 – There is a ) missing on this line.

Figures 5 through 8 – One suggestion the authors might want to consider is to change the colors of the bars and use warm colors (e.g. red, orange, yellow, or shades of red) for the single Alter data and cool colors (blue, green, purple or shades of blue) for the unshielded data. This would really make the differences between the two data sets
much more obvious and quickly draw the reader's attention to the points you make in the manuscript.

---

## Author Comment (AC1) · 20 Dec 2019

**Author's Response to Anonymous Referee #1**

We would like to thank Referee #1 for taking the time to review and offer feedback on this manuscript. Our responses to the comments and actions taken are listed below in red.

**RC:** P2, L15: In this paragraph, a short comment of how these stations are maintained after snowfall, or when they might get ice around, would be beneficial.

**AC:** This paragraph is more of an introduction to the SPICE sites and intercomparison period so a discussion of site maintenance may not be appropriate here. However, to answer the question, site maintenance was the responsibility of the site host and many sites kept service logs that were made available to improve data quality control. These logs usually note gauge servicing and instrument malfunctions.

**Action:** The following sentence has been added to the Methods sections to clarify how site maintenance issues were identified and handled in data quality control: "Where available, service logs were provided by site hosts to assist in data quality control and the identification of outliers due to servicing (e.g. rapid drops or increases in precipitation gauge bucket weights) or maintenance (e.g. instrument malfunctions or other human interventions that may impact the data)."

**RC:** P2, L28: What is a Precipitation detector?

**AC:** For SPICE, this was typically an optical disdrometer that is capable of detecting the occurrence of even light precipitation events at relative high temporal resolution (1-min in this case).

**Action:** Added the statement to describe "precipitation detector" as "typically an optical disdrometer capable of identifying the occurrence of even light precipitation.

**RC:** P3, L6: A reference after "demonstrating that the unshielded catch from both SUT types were very similar." would help in this sentence.

**AC:** This statement appears in the same sentence that references Kochendorfer et al. (2017b). The same reference applies to the latter statement in the sentence.

**Action:** To clarify, the sentence was changed to read "The transfer functions presented in K2017b were developed…, after the authors demonstrated that the unshielded catch…"

**RC:** P4, L9-11: I think that a more detailed explanation of this sentence would help to complete the idea. For example, answering what makes these evaluations "marginally independent"?

**AC:** We will modify this paragraph to clarify.

**Action:** The paragraph now states: "As discussed above, the data from all eight sites in Table 1, including data from multiple gauges of the same configuration at each of the sites, were combined to fit the transfer function models. This model, developed by pooling the data from multiple sites, was then applied to each individual gauge at each site. By applying a model developed from data collected at

multiple sites to individual gauges at each individual site, the authors maintained some independence between the model development and the evaluation."

**RC:** P4, L17: I would include this sentence as the main result in the abstract.

**Action:** The relevant sentence in the abstract now reads: "Due to the short intercomparison period, the dataset was not sufficiently large to develop and evaluate transfer functions using independent precipitation measurements, although on average the adjustments were effective at reducing the bias in unshielded gauges from -33.4% to 1.1%."

**RC:** P4, L29: What do you mean with "more natural circumstances"?

**AC:** We meant in situations more typical of those experienced by a user collecting data in the field. We will revise this sentence to reflect this.

**Action:** Sentence was changed to say: "The methodology used during SPICE for developing and evaluating the transfer functions used only a subset of the observed data (the SEDS), and although this was a robust methodology for developing transfer functions, it did not provide a comprehensive evaluation of the adjustments under circumstances more typical of users collecting precipitation data in the field where the data is less filtered to remove smaller amounts."

**RC:** P4, L36: This has to be consistent with P4, L9-11.

**AC:** We think that the revision on P4, L9-11 now sufficiently explains the issue such that no further action is required here.

**Action:** none taken

**RC:** P5, L3: I think the word "continued" should be in present tense.

**AC:** agreed

**Action:** "continued" change to "continues"

**RC:** P5, L7: I suggest completing the end of the sentence as: "...it will also isolate the performance to adjust snow measurements"

**AC:** agreed

**Action:** sentence modified as suggested

**RC:** P9, L15: I have a question here, why not using a standardized RMSE value, as it highly depends on the precipitation rates?

**AC:** There is advantages and disadvantaged to either. Standardizing the RMSE by using precipitation rate would allow better intercomparison between sites but information regarding the absolute magnitude of the error at each individual site would be lost. The most important interpretation here is the change (or

lack of) in RMSE with adjustment. Therefore, we think that using RMSE and cautioning the interpretation between sites, combined with the use of the other metrics, is the appropriate action.

**Action:** none taken

**RC:** P10, L1: Could you please add a number or a percentage to quantify the adjective"insubstantial"?

**Action:** We quantified this as < 0.005 mm for the single-Alter and as 0.05 mm for the unshielded gauges.

**RC:** P9, L11-15; P10, L5 L7-8; P10, L28-32:  these sentences describe the statistical parameters used to analyze the data. I suggest moving them to the methodology section.

**AC:** Agreed

**Action:** The description and justification of the metrics have been moved to section 2.2 Performance metrics under Methods

**RC:** P12, L10-11, The number of events that are not captured by the single-Alter shielded in XBK is over 50%, do you have any comments to propose a better monitoring system in this location?  Or how we could possibly integrate these cases to the adjusted precipitation?

**AC:** Fortunately, many of those event are small events which reduces the impact on total precipitation. However, they are still significant given the percentage of the reference precipitation amount not measured by the non-reference gauges. Currently, the most effective way to reduce this error is to increase the catch efficiency of the non-reference gauge by adding shielding. This is captured in the sentence: "Since more shielding (e.g. double-Alter) generally means a higher catch (Watson et al., 2008; Smith, 2009; Rasmussen et al., 2012; Kochendorfer et al., 2017b), more shielding would also reduce the number of unmeasured events." The use of disdrometers could also increase detection of small events in cold and windy conditons.

**Action:** Added the following sentence: "Perhaps another option to increase detection of small events in cold and windy locations would be the use of optical disdrometers paired with the conventional accumulating gauges."

**RC:** Table 1; Please, specify the meaning of "gh" in the table caption, or as a footnote.

**Action:** done

**RC:** Figures 3 and 4, I suggest increasing the panel size and/or font size.

**AC:** That's a good suggestion and we will make sure those are larger for final production. For review, we wanted to try to get them all on the same page.

---

## Author Comment (AC2) · 23 Jan 2020

**Author's Response to Anonymous Referee #2**

We would like to thank Referee #2 for their thorough review of this manuscript. We appreciate the detailed comments and suggestions. Our responses to the comments and actions taken are listed below in red.

**Major comments:**

**RC:** In the abstract and section 1.1, the authors discuss using liquid, mixed phase and solid precipitation. A possible useful addition to Table 1 would be statistics on the percentage of data that was liquid or mixed phase precipitation and the percentage of the data that was solid precipitation for each site. This could potentially help explain some of the results you see in the other tables when comparing all precipitation phases to snow only.

**AC:** We agree that this could be useful information. We calculated the percentage of precipitation events for all phases (solid/mixed/rain) for the two seasons combined that were used in this analysis using the temperature thresholds noted in this paper and in Kochendorfer et al. (2017b). They are as follows:

| Site | % Precip Events by Phase solid/mixed/rain (%) | % Total Precip by Phase solid/mixed/rain (%) | Approximate Change in RTC from All to Snow |
|------|-----------------------------------------------|----------------------------------------------|--------------------------------------------|
| XBK  | 59/16/25                                      | 43/16/40                                     | -10%                                       |
| CAR  | 39/32/29                                      | 28/33/39                                     | +3%                                        |
| CCR  | 91/8/1                                        | 85/15/1                                      | 0%                                         |
| FOR  | 31/43/26                                      | 28/49/23                                     | -20%                                       |
| HKL  | 47/38/15                                      | 44/35/21                                     | -12%                                       |
| MAR  | 44/33/23                                      | 32/45/23                                     | +12%                                       |
| SOD  | 54/39/8                                       | 49/42/8                                      | +2%                                        |
| WFJ  | 79/19/2                                       | 78/21/1                                      | +2%                                        |

There were really no surprises in these numbers. We believe what the referee was alluding to in their comment was that we should expect that the biggest changes in Relative Total Catch (from all precip to solid precipitation) should occur at sites with the smaller percentage of solid precip. This would support the hypothesis that the higher total catch of rain and mixed events is masking the errors associated with adjusting only solid precipitation. The largest drop in RTC does occur at FMG (~-20%) which also has the lowest percentage of solid precipitation events during the season (31%). However, the next highest drops in RTC occur at HKL (~-12%) and XBK (~-10%) and both of these sites exhibit a higher percentage of solid precipitation events during the study period (47% and 59% respectively). Alternatively, we could look at the precipitation totals for each type (second column in table above). Although the percentage value for total snow is generally lower than for event occurrence, it doesn't change the interpretation.

The relative amount of snow as compared to rain/mixed could possibly contribute to the difference in the statistics for FOR, but doesn't explain the differences at the other sites. We believe that the addition of a statement in the discussion referring to this will not improve the clarity of interpretation.

**Action:** Added columns in Table 1 to show the relative total precipitation by phase for each site. Added a reference to this information in Section 2.2.

**RC:** The authors show analysis using the wind data collected from 10m heights and gauge height and discuss many issues related to measuring the wind speeds.  In the discussion starting on page 13, line 20, as well as the first part of the conclusions section, no mention is made of the possible impacts of wind variations over the 30-minute averaging period.  For the windier sites used in the study, how variable were the winds over the 30 minutes and what role might that have played in some of the results?  I agree with the discussions and results the authors have expressed regarding some of the other wind-related issues but this should also be addressed.

**AC:** This very issue was addressed in previous work during the WMO-SPICE project and was published both in Wolff et al. (2015) and Nitu et al. (2018). Wolff et al. looked at variations in both 30-minute temperature and wind speed at HKL (one of the windier SPICE sites) and concluded that removing points with large variability within the 30-minute period did not significantly change the results.

**Action:** The following sentences were added to the end of the discussion paragraph that formerly started on page 13, line 20: "Additional uncertainty related to wind speed may be attributed to the variability within the 30-min mean period. Although this wasn't included in the current analysis, previous work by Wolff et al. (2015) and Nitu et al. (2018) at HKL showed that the impact of high frequency variability in the wind speed over 30-min periods on transfer functions was negligible."

 **RC:** Regarding the GEONOR gauges used in the study, the authors state they were GEONOR T-200B3 gauges.   I believe this implies a three-wire GEONOR gauge.   If that is the case, how were the data from the three wires used?  Were they averaged? This should be discussed and/or clarified in the introduction or methods sections.

**AC:** The three Geonor wires were averaged before filtering.

**Action:** The methodology was revised to reflect this

**RC:** With regards to Fig 1, equation 2 shows a collection efficiency of 1.1 at a wind speed of 0, but I don't see a clear explanation for this.  There is some discussion on P4, L1 that this might be indicative of rain at air temps > 2C, but equation 2 does not use temperature so I'm a bit confused as to why temperature is mentioned there and why eqn 2 would result in a catch efficiency > 1 at 0 m/s. Some clarification and additional explanation would be useful here.

**AC:** This is a good question and should have been addressed in the paper. Simply, the Equation 2 catch efficiency was greater than 1.0 because of the specific empirical fit to the WMO-SPICE catch efficiency data that produced the smallest error. It is due to the shape of the data that the equation was fit to, rather than a physical cause. This function was not originally published in this present manuscript, but

rather in K2017B. For this analysis, any catch efficiency that was calculated as > 1 was automatically reset to 1. This occurred relatively infrequently, since the mean wind speed during precipitation has to be less than 0.5 m s$^{-1}$ and therefore did not significantly affect the results.

The discussion on P4, L1 does not explain why the catch efficiency for Eq. 2 is greater than 1.0. This text simply explains how liquid precipitation, classified as precipitation occurring when Tair > 2 deg C, should be handled according to K2017b.

**Action:** The following has been added to the manuscript in the methodology section (2.2): "Figure 2 suggests that Eq. 2 can exceed a catch efficiency of 1 at low wind speeds. There is no obvious physical explanation for the portion of the Eq. 2 catch efficiency function (originally published by K2017b) that is > 1.0 and this is related to the empirical fit of the catch efficiency curve to the original SPICE data. For this current assessment, calculated catch efficiencies > 1 were infrequent and occurrences were automatically set to 1."

**RC:** In section 3.2 and 3.3, there is mention of potential shadowing of the wind sensor at the MAR site. Was anyone at the site contacted and asked for pictures of the sensor at the site to confirm this (assuming none of the authors have visited or work at this site)? This would seem to be an easy thing to do without supposition.  Could this also be a calibration issue with the wind sensor at that site?

**AC:** Co-author J. Kochendorfer is quite familiar with the MAR site and has done extensive work during the SPICE project on the wind measurements. This was outlined in Kochendorfer et al. 2017a. During the development of the Kochendorfer et al. (2017a) transfer functions, the ratio of the 30-min gauge-height wind speed to the 30-min 10 m height wind speed was plotted as a function of wind direction, and it was quite clear that the gauge-height anemometer was affected by obstacles in some wind sectors.

**Action:** P8, L31 has been changed from, "wind speed measurements may be shadowed…", to, "wind speed measurements were shadowed…". This statement was not based on supposition.

**Minor comments:**

**RC:** Throughout the manuscript: There is inconsistent use of the hyphen in the phrase "single Alter shielded".  In some places, there are two hyphens, in others, just one.  I believe it should be "single Alter-shielded".

**Action:** The wording in the text should now consistently be "single Alter-shielded"

**RC:** P1, L17 – Intercomparison is hyphenated here but nowhere else.  The official SPICE title appears to not have the hyphen.

**AC:** Inter-Comparison was supposed to have been hyphenated so that the acronym SPICE made more sense, but it appears to have been dropped in the literature along the way. This should be consistent

**Action:** updated the text without the hyphen in Intercomparison.

**RC:** P1, L20 – The term "windshield" is typically used to describe the front glass on a car. In the case of precipitation measurement, I believe "wind shield" is more commonly used.

**AC:** agreed

**Action:** changed the text so that "wind shield" is consistently used.

**RC:** P1, L25-27, The sentence on these lines is somewhat awkwardly written.

**Action:** Changed to "Performance is assessed in terms of relative total catch (RTC), root mean square error (RMSE), Pearson correlation (r), and Nash-Sutcliffe Efficiency (NSE). Metrics are reported for combined precipitation types, and for snow only."

**RC:** P1, L29 – What is meant by gauge configuration?

**AC:** This refers to the gauge model and the wind shield type (if any) that the gauge is paired with.

**Action:** clarified this in the text by adding "…and gauge configuration *(gauge and wind shield type)*."

**RC:** P5, L24 – This sentence reads a bit awkwardly and may just be missing a comma. Perhaps revise into two sentences.

**Action:** These sentences now read: "The algorithm removes random and systematic diurnal noise, but does not account for signal drift (an example of signal drift is a decrease in weight that occurs due to evaporation of water from the gauge bucket)."

**RC:** P6, paragraph 2 – How often were the temp/wind speed data missing?

**AC:** We can qualitatively say that the number of missing temperature and wind speed data were low, and generally the missing temperature and wind speed data coincided with missing precipitation data. Since missing data is not included in the analysis, this is largely inconsequential.

**Action:** none taken

**RC:** P6, L34 – there is an extra comma on this line

**Action:** fixed

**RC:** P6, L35 – there shouldn't be a comma after "snow".

**Action:** fixed

**RC:** P7, L5 – The term "both" is not necessary in this sentence.

**Action:** fixed

**RC:** P8, L6 – The term "both" is not necessary here either.

**Action:** fixed

**RC:** P11, L18 – season should be plural.

**Action:** fixed

**RC:** P11, L20 – I think you meant to say compounding instead of confounding.

**Action:** changed this to "complex".

**RC:** P12, L16 – There is a ) missing on this line.

**Action:** fixed

**RC:** Figures 5 through 8 – One suggestion the authors might want to consider is to change the colors of the bars and use warm colors (e.g. red, orange, yellow, or shades of red) for the single Alter data and cool colors (blue, green, purple or shades of blue) for the unshielded data.  This would really make the differences between the two data sets much more obvious and quickly draw the reader's attention to the points you make in the manuscript

**AC:** Agreed.

**Action:** We changed the colour of the bars so that the shielded gauges are shown in grey tones while the unshielded gauges are shown in blue tones. We also made some changes to help with discerning the bars if printed in grey scales.

---

## Author Response (AR2)

The following is to address the Feb 10 2020 Editor's Comments

1) General comment: the structure is not always clear, there are methods in the intro and also in the result section (e.g., section 3.4), please check that the methods are all in the methods section and so on …

AC: The authors agree that there is a brief introduction to the methods used in this study in **Section 1.3 Motivation for the extended evaluation**. Although we think that some very brief discussion on current methods is important here to clearly state how this study differs from K2017b in both objectives and methodology, we have significantly reduced this to improve the structure of the paper. The methods discussion in section 3.4 was removed. This discussion had previously been moved to the Methods section, but was inadvertently not removed from the Results section. We appreciate that the error was found.

Action: Shortened the paragraph (P4, L 30 - P5, L3) to reduce the discussion on current methodology and move the paragraph (P5, L5-11) in its entirety to the beginning of the Methods section. In Section 3.4, P11, L 15-20 were removed.

2) P1L26 and elsewhere: the term verification (establishing the truth) is not suitable here, validation would be better (or confirmation/testing)

AC: We agree. "Validation", "Assessment", or "Evaluation" are all more appropriate

Action: Occurrences of "verification" have been replaced with something more appropriate.

3) P1L33: 102% and 123% would be clearer here (a few lines above you use 123% to refer to an increase)

AC: We agree that this should be consistently stated as a percentage of the reference. However, this needs to be clearly stated so that the reader knows that this is total catch relative to the reference rather than a reported bias.

Action: This sentence now reads "Generally, windier sites such as Haukeliseter (Norway) and Bratt's Lake (Canada) exhibit a net under-adjustment (RTC of 54% to 83%), while the less windy sites such as Sodankylä (Finland) and Caribou Creek (Canada) exhibit a net over-adjustment (RTC of 102% to 123%)."

4) Eq 1&2. Please clarify whether other functions had been considered in K2017b

AC: From Kochendorfer et al. (2017a) and personal communication, other functions were considered, such as the sigmoid function published by Wolff et al. (2015). The decision by Kochendorfer to use the current function was based on previous studies by Goodison (1978) and

Yang et al. (1999). It was stated in Kochendorfer et al. (2017b) that the more complex sigmoidal function produced similar biases and RMSE as the more simple functions, so the more simple option was chosen.

Action: The following sentence was added to P4 after L3: "Kochendorfer et al. (2017a, 2017b) examined the use of more complex transfer function forms for adjusting solid precipitation measurements, such as the sigmoid function used by Wolff et al. (2015), but found that the simpler forms had similar bias and RMSE characteristics as the more complex forms."

5) Could you add a photo from each of the sites (also showing the surroundings)?

AC: We think that the addition of 8 or more photos in this paper would significantly add to the length and detract from the readability. It would also be quite difficult to illustrate the site surroundings with only a single photo per site. Rather, we refer the reader to either the SPICE report or the SPICE site commissioning reports, both available online. These references provide more detail about site layouts and exposure than is possible in this paper.

Action: Although P2 L20-23 states that "A detailed layout and site description for each of these eight sites can be found in the WMO-SPICE site 20 commissioning reports at: http://www.wmo.int/pages/prog/www/IMOP/intercomparisons/SPICE/SPICE.html and in the WMO-SPICE final report (IMO 131 found at http://www.wmo.int/pages/prog/www/IMOP/publications-IOM-series.html; Nitu et al., 2018).", we added a note that site photographs are also available via these documents.

6) P3L29: were filtered (data is pl)

Action: corrected

7) P3L36: why symmetric around zero degrees, the snow/rain threshold usually is above zero degree

AC: The phase change thresholds are somewhat arbitrary but based on previous (pre-SPICE) analysis that was documented and adopted during SPICE (Nitu et al., 2018; Kochendorfer et al., 2017a, 2017b, 2018). Although we recognize that phase changes can occur outside of these ranges, the thresholds were set to be relatively conservative and to factor in the rain to snow transitional mixed phase. The recommendations on the thresholds were based on analysis by Smith (2008, 2009) and Watson et al. (2008) and other undocumented results using manual observations collected during the previous WMO intercomparison (Goodison et al., 1998). The decision for SPICE thresholds was based on analysis by Wolff et al. (2015) at HKL, which distinctly showed changes in the shape, slope, and scatter of transfer function curves as the phase changed through the ranges of the current thresholds (as confirmed by present weather sensors), and by analysis conducted by Kochendorfer et al. (2017a) for MAR using a similar

methodology. Furthermore, we mention in our discussion that there are alternate approaches to phase discrimination (i.e. Harder & Pomeroy, 2013) that could be explored for this type of analysis, but this is out of scope for this paper.

Action: Added the sentence: "The temperature thresholds for phase discrimination in the SPICE analysis are based on disdrometer measurements of precipitation type from Wolff et al. (2015) and Kochendorfer et al. (2017a)." to P3 L36.

8) There are a lot of abbreviations, are really all of them needed?

AC: We are unsure as to what abbreviations this comment refers to. In the paper, we abbreviate the site names (to avoid using Weissfluhjoch and Haukeliseter repeatedly) and to reduce the table widths. We use abbreviations such as SEDS and DFAR to be consistent with previous usages. We abbreviate Kochendorfer et al. (2017b) to K2017b as this paper is cited frequently, as required, since much of the groundwork for this current paper was established in the cited paper. We believe that all abbreviations are well defined and assist the flow of the paper.

9) P5L11: isolate? Other word? Unclear what is meant

AC: Perhaps "segregate" would have been a better word. It simply refers to the process by which the solid precipitation is separated from other precipitation phases and assessed separately.

Action: Rephrased as: "Extending the K2017b evaluation, this assessment will consider transfer function performance using both: (1) data over the entire winter season, regardless of precipitation phase; and (2) solid precipitation measurements only, to focus on the most challenging and critical adjustments."

10) P11L17: "NSE has a higher sensitivity to bias and outliers than (r)", note that r has no sensitivity at all to biases, and can you explain why NSE is less sensitive to outliers? What does an NSE=0 imply in this case (compare as good as mean runoff for runoff simulations)

AC: In our opinion, we are not convinced that the addition of the NSE analysis significantly contributes to the results of this paper. Generally, it is not used for these types of intercomparisons and unfortunately, the co-authors are not entirely versed on using this metric in practice. However, the following is our response to the comments:

The sentence in question was placed there to justify the use of NSE to the reviewers. According to Wikipedia (for example) (https://en.wikipedia.org/wiki/Nash%E2%80%93Sutcliffe_model_efficiency_coefficient), NSE is used to assess the predictive power of hydrological models and NSE=0 indicates that the model predications are as accurate as the mean of the observed data. In this context, an NSE value of

zero indicates that the adjusted measurements and unadjusted measurements are similarly distributed relative to the DFAR measurements. Regarding the comment about sensitivity to outliers, it is our understanding that NSE is more sensitive to outliers than the Pearson correlation coefficient, on account of the variance term in the denominator of the former.

Note that the sentence in question has also been moved to Section 2.2.

Action: Modified the sentence to read "The measure has a higher sensitivity to outliers than the Pearson Correlation coefficient and this can have both advantages and disadvantages in this assessment."

11) P11L22: Unadjusted NSE: I assume you mean NSE for unadjusted precip measurements. As there are adjusted correlation coefficients and NSE, your formulation (also post-adjusted NSE in L25) are misleading

AC: agreed

Action: The affected sentences on P11 L22-25 have been revised to make the statements less misleading.

[revised manuscript text omitted]

---

## Author Response (AR3)

**Authors' response to Editor's comments**

**June 24, 2020**

**Editor's Comments:**

Thanks for your revisions. While I am happy with most of them, I am less convinced by the responses regarding NSE. Going back, I realize that NSE was included based on my initial comment. What I was concerned at that time, was that using only Pearson r could result in misleading results as a perfect value of one actually could be achieved with large errors. For streamflow this problem is usually solved by using NSE (or KGE) instead of Pearson and, thus, I suggested to do something similar in your study. However, precip data are different than streamflow and NSE might not be the best metric. If you keep using NSE you need to address the issues I raised in the previous round. Alternatively, you might use another metric. I am sorry if there was a misunderstanding, I did never mean to force you to use NSE, but I wanted you to use a metric where the perfect value (of one) ensures that the fit is perfect (and without any bias).

**Author's Comments:**

It is agreed that the  interpretation of NSE in this context was unclear. After deliberation with the editors and co-authors, it was decided to replace NSE with another metric employed in previous transfer function evaluations, the "Percentage of Events" or PE. This metric was used by Kochendorfer et al. (2017b) to complement the other assessment metrics used in their analysis. The PE is defined as the percentage of 30-minute precipitation measurements made by the non-reference gauge that have an amount within 0.1 mm of the concurrent reference measurement.  Perfect agreement between the 30-min measurements made by the non-reference gauge and the reference, within the pre-determined measurement uncertainty of the reference gauge (0.1 mm), would produce a PE of 100%. PE demonstrates how the adjustments impact the bias and the uncertainty of individual precipitation events (30-min precipitation measurements, in this case), providing additional perspective on trends observed using the other metrics.

In theory, unadjusted precipitation measurements from non-reference gauge configurations would have relatively high systematic bias (> 0.1 mm), due to wind undercatch, as compared to the reference, and hence have a lower PE. After adjustment with a transfer function, the number of measurements within 0.1 mm of the reference should increase, thereby producing a higher PE. In some circumstances, however, a transfer function adjustment could increase the magnitude of gauge observations that are already within the 0.1 mm threshold, contributing to a decrease in the number of events agreeing with the reference and thereby decreasing PE.

The removal of the NSE results and the addition of the PE results did not change any of the conclusions of the manuscript; however, the inclusion of PE allowed for improved contextualization of results for the other metrics.

The assessment of transfer function adjustments of precipitation gauge measurements is complex, as it involves consideration of bias and uncertainty in precipitation measurements over different time frames (seasonal or fixed-duration events). It is challenging to find a single statistical metric that captures all of these aspects. Accordingly, we have employed complementary metrics in this regard, and have highlighted the focus of each metric in the revised manuscript.

**Actions:**

Abstract, Page 1: The abstract has been updated to state the general results of the PE interpretation rather than the NSE metric.

Section 2.1, Page 7: The description of the evaluation methods (lines 12-23) were moved to Section 2.2 (see following note) to improve the flow of the manuscript and reduce redundancy.

Section 2.2, Page 7: The section header was changed from "Performance metrics" to "Performance assessment" to better reflect the contents of the section.

Section 2.2, Page 8: The description and justification of the NSE statistic was removed and the description and justification of the PE statistic was added (lines 12-18). Some revision to the description of the other statistics were added to highlight how these metrics are complementary.

Section 3.4, Page 12: The previous results related to the NSE metric have been removed and the results associated with the PE statistic have been added (lines 12-22).

Section 4, Page 13: The discussion focusing on the NSE results have been removed and the interpretation of the PE results has been added, for the most part occurring in lines 18-33. Some other minor edits were performed on Pages 14 and 15 to update the discussion for PE.

Section 5: Only two small revisions were made to the conclusions sections, updating for the replacement of NSE with PE (line 11 and line 18).

Tables 8 and 9, Pages 30-31: Tables were updated to show PE results, deleting NSE results

Figure 8, Page 38: replaced the NSE bar graph with the PE bar graph.

[revised manuscript text omitted]

---

## Author Response (AR4)

**Authors' response to Editor's comments**

**July 8, 2020**

**Editor's Comments:**

Thanks for these revisions. While I honestly saw here a chance for a more creative performance measure, exchanging the NSE for PE at least removes the issue of using NSE with precip as discussed before. Judging the values of PE might be a bit tricky and using some benchmark values to make results more comparable would be valuable, but I leave this to the authors to decide.
What I would strongly recommend, however, would be adding an equation to better describe the PE statistic.

**Author's Comments:**

The Editor's comments and decision to accept the manuscript for publication are appreciated. As recommended, we added an equation that better describes the PE statistic.

**Actions:**

Starting on Page 7 Line 29, we revised the description of the PE statistics, including the addition of Equation 3

[revised manuscript text omitted]